# Perivascular Mesenchymal Stem/Stromal Cells, an Immune Privileged Niche for Viruses?

**DOI:** 10.3390/ijms23148038

**Published:** 2022-07-21

**Authors:** Grégorie Lebeau, Franck Ah-Pine, Matthieu Daniel, Yosra Bedoui, Damien Vagner, Etienne Frumence, Philippe Gasque

**Affiliations:** 1Unité de Recherche en Pharmaco-Immunologie (UR-EPI), Université et CHU de La Réunion, 97400 Saint-Denis, France; greg.lebeau@live.fr (G.L.); franck.ahpine@gmail.com (F.A.-P.); matthieu.daniel2309@gmail.com (M.D.); yosra.bedoui.bouhouch@gmail.com (Y.B.); etienne.frum@gmail.com (E.F.); 2Laboratoire d’Immunologie Clinique et Expérimentale de la ZOI (LICE-OI), Pôle de Biologie, CHU de La Réunion, 97400 Saint-Denis, France; 3Service Anatomo-Pathologie, CHU de la Réunion, 97400 Saint-Denis, France; 4Service de Médecine Interne, CHU de la Réunion, 97400 Saint-Denis, France; damien.vag@gmail.com

**Keywords:** immunity, mesenchymal stem cells, stromal cells, fibroblast, pericytes, virus, persistence, chronic inflammation, innate immunity, immune-regulation, neural crest, COVID-19, chikungunya

## Abstract

Mesenchymal stem cells (MSCs) play a critical role in response to stress such as infection. They initiate the removal of cell debris, exert major immunoregulatory activities, control pathogens, and lead to a remodeling/scarring phase. Thus, host-derived ‘danger’ factors released from damaged/infected cells (called alarmins, e.g., HMGB1, ATP, DNA) as well as pathogen-associated molecular patterns (LPS, single strand RNA) can activate MSCs located in the parenchyma and around vessels to upregulate the expression of growth factors and chemoattractant molecules that influence immune cell recruitment and stem cell mobilization. MSC, in an ultimate contribution to tissue repair, may also directly trans- or de-differentiate into specific cellular phenotypes such as osteoblasts, chondrocytes, lipofibroblasts, myofibroblasts, Schwann cells, and they may somehow recapitulate their neural crest embryonic origin. Failure to terminate such repair processes induces pathological scarring, termed fibrosis, or vascular calcification. Interestingly, many viruses and particularly those associated to chronic infection and inflammation may hijack and polarize MSC’s immune regulatory activities. Several reports argue that MSC may constitute immune privileged sanctuaries for viruses and contributing to long-lasting effects posing infectious challenges, such as viruses rebounding in immunocompromised patients or following regenerative medicine therapies using MSC. We will herein review the capacity of several viruses not only to infect but also to polarize directly or indirectly the functions of MSC (immunoregulation, differentiation potential, and tissue repair) in clinical settings.

## 1. Introduction

Mesenchymal stem cells (MSCs) are a subset of non-hematopoietic stem cells found at low frequency, mainly located around vessels (hence also named pericytes) in resting conditions but with high proliferation and multilineage differentiation capacities to orchestrate tissue repair mechanisms [1,2,3,4]. MSCs can be isolated from a wide variety of organs, including bone, lung, adipose tissue, umbilical cord, placenta, brain, liver, kidney, and synovial joints [5,6]. Their identification is facilitated by the use of selective cell markers (e.g., CD90/Thy1, CD140b/PDGFR, CD146/Muc18, CD271/Low affinity NGFR, or CD248/Endosialin/Tumor endothelium marker 1) [7,8,9] and the lack of expression of canonical hematopoietic cell markers such as CD3, CD19, and CD14 for T, B and monocytes, respectively [4,10]. MSC can be derived from mesoderm and also from neural crest (NC) embryonic tissues, and this may explain their capacity to express neuroglial/Schwann cell markers such as glial fibrillary acidic protein (GFAP) and myelin P zero (MPZ) in vitro and in situ [4,7,11,12,13,14,15,16,17,18,19].

In the last few years, MSCs have attracted growing interest in the research community [20]. Indeed, MSCs play a pivotal role in tissue maintenance in response to injury due to their major immunoregulatory activities, avoiding autoimmunity, and the control of the inflammatory response. Hence, MSCs have important therapeutic potentials [2,21].

However, a particular attention must be paid to MSC’s dark side given that they are at the core of the fibrosis response in the case of non-resolving injuries [22,23,24]. Indeed, reactive MSCs, for instance exposed to high levels of TGF-beta 1 (TGF-β1), can differentiate into myofibroblasts, which are known to produce excessive amount of extracellular matrix proteins (ECM) such as collagen 1 and fibronectin [9]. This fibrosis response initially contributes to tissue repair but, if uncontrolled, would lead to organ dysfunctions [24]. Thus, therapeutic avenues should be explored to avoid the differentiation of MSC into myofibroblasts, which potentially leads to the development of pathological wound healings, including permanent scar tissue and loss of tissue functions. It should also be possible in already established chronic inflammatory conditions to initiate the reversion of the pathological highly proliferating myofibroblasts into a more quiescent MSC. Interestingly, several drugs have shown such therapeutic potentials and including the anti-diabetic biguanide drug metformin used recently to control lung and kidney fibrosis [25,26].

Critically, among the plethora of unresolved tissue injury conditions, infectious diseases caused by viruses are one of the most life threatening.

Perivascular MSC, closely associated to endothelial cells through a myriad of cell-cell communication cues [11,22], are strategically located at the portal of entry of viruses but little is known about their capacity, in situ, to control the infection through a local innate immune response and/or to engage innate as well as T/B adaptive immunities. Moreover, the capacity of viruses to impact directly MSC’s functions should be considered, as they tend to operate through ill-characterized mechanisms. For example, HIV infection of bone marrow (BM)-derived MSCs (BM-MSC) is associated with the inability to support hematopoietic stem cell (HSC) expansion and HIV-related cytopenia [27], while HIV infection of kidney MSC (i.e., mesangial cells) is linked to HIV-associated glomerulosclerosis [28]. Kallmeyer and colleagues have recently reviewed the role of MSC in HIV pathology; this aspect will be briefly discussed below while focusing on many more viruses known to affect MSC activities [29].

Moreover, latently infected MSCs (as it is the case for various herpesviruses) represent a risk following transplantation in immunocompromised patients [30].

The aim of the review is to raise awareness about the role of MSCs in the development of virus-related pathologies, including as a virus reservoir and virus rebound mechanism, that should be considered for the development of MSC-based therapies.

## 2. Cell Biology and Immune Functions of Mesenchymal Stem Cell (MSC)

### 2.1. Definition of MSC

The acronym MSC refers either to: Mesenchymal Stem Cell, a term popularized by Caplan in the 1990s [1] and broadly used in the past decades, or Multipotent Mesenchymal Stromal Cell, which is the terminology promoted by Mesenchymal and Tissue Stem Cell Committee of the International Society of Cell Therapy [31].

The adjective “mesenchymal” (“middle” in Greek) suggests a putative mesodermal origin of MSCs, implying that MSCs may derive from the “middle” layer during the embryonic development [1]. A neuroectodermal NC origin of MSC has been highlighted in many tissues by several investigators [4,12,15,16,17,32,33,34]. NC cells have the capacity to migrate and to participate in the organization of ectodermal and endodermal tissues [33]. This migration ability may be retained during adulthood, by homing to injured tissues [1,6,20,35,36]. Even if MSCs terminology and definition have been a matter of debate (notably about their “stemness”), there is a consensus on certain elements.

First, MSCs of the BM are distinguishable from hematopoietic stem cells, because of their ability to adhere to plastic vessel and to grow [1,31,37,38,39]. Morphologically, MSCs are spindle-shaped cells that after several passages bear a more homogeneously fibroblastic phenotype [40].

Second, MSCs are characterized by a singular expression of surface proteins. Among the classical markers of MSCs, we aforementioned CD90 (or Thy-1), CD105 (or endoglin, TGF beta receptor), and CD73 (or ecto-5′-nucleotidase) [8,37,38,41]. Other markers have been described for MSC such as CD140b (or PDGFR beta), CD271 (or low-affinity NGFR) [8,42,43], CD146 (MUC18), and CD248 (or endosialin/tumor endothelium marker 1) [7,9,22]. However, these markers are not specific to MSCs and can be found on other cell types. Additionally, MSCs are negative for CD45 (*pan*-leukocyte marker) and CD34 [37,38], even if some investigators evoked that CD34 may be expressed but lost in ex vivo culture expanded MSC [8,44]. Actually, it is important to note that the in vitro expression of some markers does not always correlate with their expression patterns in vivo [40].

Third, MSCs are multipotent progenitor cells, able to differentiate at least in vitro into three different subsets: adipocyte, chondrocyte and osteoblast [6,37,38]. In addition to these cell types, other possibilities of differentiation have been evoked [45]. Pericytes in the arteries can acquire a macrophage-like phenotype even with phagocytic properties [46,47]. This phenotype switching involves the transcription factor Krüppel-like factor 4 (KLF4) and has been associated with development of atherosclerosis. Some culture conditions can promote smooth muscle and striated muscle gene expression, whereas others promote cardiac or liver gene expression [48]. Finally, glial/neuronal differentiation potential has also been reported [40,41] and which may be due to the NC origin of specific subsets of MSC [7,11,12,15,16,17,19].

MSCs have been isolated from various tissues [4,5,6,40] and including: bone-marrow [49], adipose tissue [50,51], lungs [52], synovial membrane [53], kidney [6], liver [54], dental tissue [55,56,57,58,59], cord blood [60,61], and amniotic fluid notably [62]. We highlight that different names have been attributed to these cells depending on their tissue locations (Table 1).

### 2.2. Origins of Pericytes/Perivascular MSC-Fibroblasts Derived from the Neural Crest and/or Mesoderm Embryonic Tissues

The ontogeny of MSC before they rich their final position in adult tissues is still a matter of debate [4]. The identification of MSC relies on the characterization of genetic and protein markers (e.g., tyrosine kinase PDGF α or β receptors, Schwann cell myelin P zero, glioma-associated transcription factor Gli1, collagen, Acta2/alpha SMA, Cspg4/Neuronglial 2-NG2, CD146/Melanoma cell adhesion molecule, CD248/Tumor Endothelial marker 1-TEM1) not restricted to MSC but also shared with pericytes (first identified by the French scientist Charles Rouget in 1873), vascular smooth muscle cells (VSMC), and perivascular fibroblasts [4,5,6,9,11,22,71]. Of note, the latter cell subset do not really fit the definition of pericyte because they are not embedded in the vascular basement membrane [72].

It is now generally accepted that a large pool of MSC is found essentially at the perivascular level and with morphology and marker expression profile similar to pericytes [5,73]. Several studies have shown that post-capillary venule pericytes from the bone marrow are able to differentiate into differentiated MSCs such as osteoblasts and chondrocytes in vivo [74]. More recent genetic lineage-tracing experiments and single-cell RNA sequencing data has reinforced a close link between pericytes and MSC phenotypes, particularly in the CNS, a tissue where the highest density of pericytes has been found in the body. RNA profiling of mouse brain vasculature revealed a rather unique pool of perivascular cells made of a two pericyte clusters and three subsets of perivascular fibroblasts [69,75]. Pathway and gene ontology enrichment analyses revealed that fibulin+ type I fibroblasts are the main subtype involved in ECM production and fibrosis. The type III (cell migration-inducing protein-CEMIP+ perivascular fibroblasts) showed robust expression of various growth factors, including VEGF-A. Interestingly, the type I to type II (potassium calcium-activated channel subfamily M alpha 1-KCNMA1+ fibroblasts) trajectory was continuous with pericyte type 2 suggesting a lineage from type I to type II to pericytes and consistent with a study in zebrafish demonstrating the stem cell potential of perivascular fibroblasts to differentiate into pericytes [76]. It was estimated that ten perivascular fibroblasts were present per intersegmented vessel but only less than 10% of these cells could differentiate into pericytes. Garcia et al. further discussed the possibility that type II perivascular fibroblasts in the brain probably represent an intermediate state exhibiting a transitional mural cell transcriptional phenotype [69].

MSC have several different developmental origins as reviewed by Majeski [77]. The majority of the MSC/pericytes in the head region, including the CNS, are neural crest (NC) derived, as demonstrated in chick-quail chimeras carried out initially by the French scientist Nicole le Douarin and colleagues [78]. In the peripheral nerves, NC will give rise to perineurial fibroblasts and Schwann cells [79]. More recently, two independent studies published in 2017 have suggested that brain pericytes could also be derived from mesoderm-derived myeloid progenitor cells [80,81]. Studies on the thymus demonstrated that perivascular MSC/pericytes are derived from the NC [32,82,83]. The origins of pericytes in the gut [84], lung [85] and liver [13,86] have been mapped to the mesothelium although NC can give rise to MSC-like cells in the gut [87].

In the kidney, the metanephric mesenchyme of the intermediate mesoderm will give rise to nephrons (from the distal convoluted tubule to the podocytes) and also to all major stromal interstitial cells, including the pericytes, perivascular fibroblasts, VSMCs, and mesangial cells [88]. a NC origin of a subset of perivascular fibroblasts of the kidney and contributing to fibrosis has been proposed from genetic lineage tracing experiments using the Schwann cell Myelin P Zero promoter-GFP/LacZ mice [12]. In the aorta, MSC may have at least four different developmental origins, secondary heart field, NC, somites, and splanchnic mesoderm. This invasion of mesothelial cells occurs at about the same time as the appearance of primitive endothelial and hematopoietic progenitors within the splanchnopleura. The primitive endothelial cells (EC) within the splanchnopleura colonize the floor of the aorta and differentiate in situ to produce the vasculature of the body wall, kidney, visceral organs, and limbs [77]. This process of vasculogenesis involving PDGF high expression by EC is consistent with the notion that mesothelial-derived MSCs are localized to BM via the invasion of the vasculature. Coronary vessels in the heart appear to have a similar development [89]. Mesothelial cells are known to undergo epithelial-to-mesenchymal transition (EMT) to delaminate and to migrate into the organs to produce their mesenchymal components. Interestingly, recent studies also point to a close ontogenic relationship between pericytes/VSMC and perivascular fibroblasts in many organs and supporting the current paradigm of such relationships in pathological settings for instance in the brain and lungs. The recent studies preach for the existence of a continuum of pericytes/perivascular MSC-fibroblasts cell phenotypes observed along vessels (and possibly nerves) and which suggest that these cells can (trans)differentiate into each other in conjunction with vessel/axonal/tissue remodeling. However, this interesting and promising paradigm requires further investigation.

The close relationship of MSC and progenitors with the vasculature will endow them as a possible source of new cells for physiological turnover for the repair or regeneration of local lesions. The canonical and current scenario is that damage to any tissue would release the MSC from its perivascular niche, they will divide and secrete immunoregulatory and trophic factors. Different signaling mechanisms may govern MSC mobilization from the perivascular niche, detaching from the endothelial cues and invading the parenchyma in response to injuries. This is exemplified by the importance of PDGF-B/PDGFRβ which has been demonstrated in many organs such heart, lung, and gut [90].

To date, we have a better idea into the embryonic origin of pericytes/perivascular MSC-fibroblasts in different organs but it is still critical to decipher the mechanisms governing their proliferation spreading along (as well evading) growing vessels in conjunction with angiogenesis (Figure 1). The capacity of these cells to circulate in the blood in various disease settings is of great and emerging importance from a clinical standpoint and including the identification of novel predictive soluble biomarkers of an ongoing pathological process in the tissues. Indeed, CD45−CD31−PDPN+ proinflammatory mesenchymal, or PRIME, cells have been identified in the blood from patients with rheumatoid arthritis, and these cells shared features of inflammatory synovial fibroblasts and predicting inflammatory flares [91]. This line of future studies is of great importance in cancer and other chronic inflammatory diseases associated with infectious diseases. In the context of cancer, some studies relate to the capacity of NG2+ pericytes to give rise to mesenchymal tumors (i.e., osteosarcoma) [92].

MSCs can migrate along nerves and vessels during development and reside in virtually all post-natal organs and tissues [6,15,16] (Figure 1). Along the nerves, MSCs are also known as non-myelinating precursor Schwann cells [19,93]. Their location around vessels to form perivascular immune privileged niches has been demonstrated by several teams [4,5,6,9,35,36,94,95,96]. Moreover, perivascular MSC may be able to sense and respond to an event (e.g., virus) in the local environment, via their ability to promote tissue immunoregulatory activities (see below) [36,44,97,98]. Perivascular MSC may therefore represent ideal tissue sanctuaries for viruses remaining dormant while protected from immune attack and rebounding particularly in the context of immunosuppressive drug treatments.

### 2.3. Role of MSC in Health (Immunoregulatory Activities and Tissue Repair) and Diseases (Fibrosis)

As described in the model of da Silva Meirelles et al. proposed in 2006, the MSCs in the vascular wall of tissue contribute to tissue maintenance (Figure 2) [6].

Their motility ability allow them to migrate to the site of inflammation or injury to establish an appropriate response [1,6,20,35,36]. This response indubitably involves their immunoregulatory activities, that have already been comprehensively reviewed by several teams [10,36,44,98,99,100,101]. Among these well-known activities, MSCs modulate innate immune response by different mechanisms: decreasing Dendritic Cells (DCs)’ antigen presentation capacity, maturation and cytokine secretion [97,102]; reducing neutrophils burst respiration [103] and apoptosis; inhibiting natural killer (NK) cell proliferation, cytotoxicity and cytokine production [104,105,106,107]; inhibiting pro-inflammatory factor secretion by activated macrophages [97,108,109,110]. MSCs also exert immunoregulatory activities on adaptive immunity by: reducing T cell proliferation [44,111]; promoting T cell shift from pro-inflammatory (Th1) to anti-inflammatory (Th2) states [97], inhibiting cytotoxic CD8+ T cells [112,113]; inhibiting B cell proliferation through direct effect and T cell-mediated inhibition [114,115,116]; inducing the proliferation of regulatory T cells (Treg) [117,118,119,120].

These actions on the immune system involve both contact-dependent and contact-independent mechanisms. Indeed, MSCs have been shown to have effects on cell survival, function and proliferation of various immune cells by directly interacting with cell surface molecules and receptors [98]. For example, MSCs were shown to inhibit both T cell and B cell proliferation by the interaction between programmed cell death protein 1 (PD1), expressed by lymphocytes, and its ligand PDL1 expressed by MSCs [111]. Additionally, Fas (CD95)—Fas ligand (CD95L) axis is engaged and can induce inflammatory T cell apoptosis [121]. MSCs have also been shown to express the molecule CD200 (named OX2 in rodents) and regulating DC as well as macrophage/microglia immune cell activation via CD200R [122,123,124,125]. Liu and colleagues provided recently a very comprehensive review on the role of many more immunoregulatory cell surface ligands and receptors (e.g., Galectin1, 3, 9) allowing MSC to control directly innate and adaptive immune cells [124].

Contact-independent mechanisms involve the release of soluble immunoactive substances and extracellular vesicles (EVs), which form the MSC-derived secretome. Prostaglandin E2 (PGE2) and indoleamine 2,3-dioxygenase (IDO) (or nitric oxide, NO in mouse) expressed by MSC have been largely described for their immunosuppressive properties, for comprehensive review [10]. The chemokine—iNOS—IDO axis mediated by MSCs leads to T cell inhibition (Shi et al. [10]). Additionally, PGE2 production by MSCs leads to IL-10 secretion by M2-polarized macrophages [110], inhibition of DCs [97], reduction of NK cell activity [105], inhibition of Th17 cells and induction of Tregs [126]. Of note, other immunomodulatory factors have been reported to be secreted by MSCs: TGF-β1, hepatic growth factor (HGF), metalloproteinase-modified CCL2 (mpCCL2), leukemia inhibitory factor (LIF), and human leukocyte antigen-G5 (HLA-G5) [44].

Moreover, mounting evidence in recent years suggests an important role of EVs containing non-coding RNAs, including miRNAs, in the regulation of the immune system by MSCs [127,128].

The recruitment of immune cells to the injury site is granted by the release of soluble factors that leads to cell chemotaxis. Of note, MSCs can participate to this phenomenon by secreting a broad range of chemokines: CCL2, CCL3, CCL4, CCL5, CCL7, CCL20, CCL26, CXC3CL1, CXCL5, CXCL11, CXCL1, CXCL2, CXCL8, CCL10, and CXCL12 [10,44]. It is important to note that immunoregulatory activities of MSC will be markedly upregulated in response to the canonical IFN-gamma produced by T and NK cells [10].

Of further note, MSCs were initially thought to be important in regenerative medicine due to their ability to differentiate into multiple cell lineages, thereby supporting tissue repair. However, studies of past decades have found that regenerative activities on injured tissue were more likely associated with the MSC-derived secretome rather than the differentiation potential of engrafted MSCs [41,100,129,130]. Indeed, during a lesion, diverse mechanisms governed by MSCs-derived secretome participate in tissue regeneration and homeostasis [36,41,44]. Among them, we can cite: (1) anti-apoptosis effects mediated either by vascular endothelial growth factor (VEGF), HGF, insulin-like growth factor-1 (IGF-I), stanniocalcin-1, TGF-β, basic fibroblast growth factor (bFGF) or granulocyte-macrophage colony-stimulating factor (GM-CSF); (2) anti-scar with bFGF and HGF; (3) support and growth of tissue progenitor cells mediated by stem cell factor (SCF), LIF, SDF1 alpha/CXCL12, macrophage colony-stimulating factor (M-CSF) and angiopoietin-1; (4) angiogenesis stimulated by bFGF, VEGF, placental growth factor (PIGF), CCL2, interleukin 6 and ECM [44]. Finally, as aforementioned MSCs are able to mediate tissue regeneration by EVs release [131,132,133,134,135]. These EVs contain notably micro-RNAs that target pathways involved in angiogenesis and tissue remodeling [130].

### 2.4. MSCs’ Pathological Contributions (Fibrosis and Vessel Calcification)

Despite the beneficial activities that would turn MSCs into an obvious therapeutic approach [2], MSCs are also involved in pathological processes such as cancer [136,137,138] or fibrogenesis (Figure 2) [9,22,23]. In order to maintain the integrity of an organ, a fibrous scar composed of collagen is formed, leading (if uncontrolled) to chronic inflammation, fibrosis and to a loss of organ function [139]. Scar-forming cells are myofibroblasts, which were at length thought to be from an epithelial origin, after epithelial-mesenchymal transition (EMT) [24,140,141]. However, mounting evidence is pointing out the role of resident perivascular MSCs in myofibroblast differentiation and expansion, for comprehensive review, [9,14]. Perivascular MSC (also named pericytes by some authors) are nowadays more and more widely evoked as the main source of collagen-producing cells in fibrosis [11,94,142,143,144,145,146], notably thanks to genetic fate tracing experiment [12,14,143,147,148]. The role of MSCs in giving rise to myofibroblasts has been validated in several models of tissue fibrosis as reviewed by El Agha and colleagues in 2017. For instance, resident perivascular MSCs (genetically traced using hedgehog transcriptional activator glioma-associated oncogene homolog 1 (Gli1)+ or MPZ+ promoters to drive fluorescent protein expression, e.g., GFP) and its profibrotic activity have been described in murine model of fibrosis in either the kidney [12,147], lungs [147,149], heart [147,150], liver [147,148] or bone marrow [151]. Table 2 summarizes and provides an updated version of the data obtained using genetic lineage tracing as well as single-cell RNA sequencing experiments.

Blood vessel remodeling can occur across a variety of pathologic conditions, including osteogenesis-like calcification (arteriosclerosis) and atheroma plaque formation (atherosclerosis) [183,184]. Although some reports have suggested that adventitial fibroblasts can contribute to pathologic changes within the vessel intima and media [185], it has now been shown, using GLI-1 line tracing experiments, that MSC can contribute to vessel calcification through a process of transdifferentiation into osteoblast-like cells [186].

The pathological role of MSCs is mostly associated with an initial and unresolved injury. Among the causes of injury, we will now focus on viral infection and how it affects resident MSCs in the different organs.

## 3. MSC Viral Infection and Host Responses

MSCs as aforementioned have mostly critical roles during injury, participating in immunomodulation and tissue repair. In addition, their perivascular location in multiple organs makes them potential targets for viral infection, as summarized in Table 3.

We will address the different viral infections that affect MSCs and the resulting innate immune response given that MSC are immunocompetent gatekeepers known to express pattern recognition receptors (PRRs, e.g., RLR, TLR) and downstream signaling molecules (e.g., NF-κB, IRF3/7) [20,99]. In parallel to macrophage polarization, two distinct subsets of MSC may exist in tissues (MSC1, proinflammatory and MSC2 anti-inflammatory) [249,250]. In response to acute tissue injury and the release of alarmins derived from cell debris (apoptotic cells), MSC have a type 1 phenotype and help to recruit lymphocytes to sites of inflammation using MIP-1a and MIP-1b, RANTES/CCL5, CXCL9, and CXCL10 and to promote the clearance of cell debris and tissue repair. If the injury is too important and associated to cell necrosis and high levels of TNF-α and IFN-β produced by monocytes and T cells, respectively, MSC adopt an immune-suppressive phenotype (MSC2) by secreting high levels of soluble immunoregulatory factors, including kynurenine (IDO pathway), PGE2 (COX2 pathway), NO, TGF-β1, IL10, Hepatocyte Growth Factor (HGF), and hemoxygenase (HO), that suppress adverse T cell proliferation and possible autoimmune response. TGF and IL10 will further control adaptive immune responses by mobilizing FoxP3+ T regulatory cells. Interestingly, viruses (e.g., those stimulating the TLR3 pathway by Poly I:C) may promote MSC2 anti-inflammatory and immunosuppressive phenotype to their own advantages by limiting the adaptive immune responses [251].

### 3.1. MSCs a Gatekeper and/or Reservoir of Viruses

#### 3.1.1. Bone Marrow-Derived MSC (BM-MSC)

BM-MSCs are among the first and best described subsets of MSCs [49]. Genetic line tracing experiments using either Nestin or GLI-1 promoters indicated that perivascular BM-MSC could be derived from NC embryonic tissues, expressed neuroglial markers, as well as the beta3 adrenergic receptors arguing for a plausible role of the sympathetic nervous system to control MSC functions [17,64,151]. Physiologically, BM-MSCs constitute a stromal cell niche for HSCs, supporting their stemness and education [17,64,252]. Due to their relative ease of access, they represent a powerful source of cells as much for the study of the properties of MSCs as for the investigation of new therapeutic avenues. Indeed, BM-MSCs are now widely used as treatment given their beneficial immunoregulatory properties. They are used as carriers for the delivery of miRs or protein factors with therapeutic activities, physiologically expressed by MSCs or induced following an adenoviral or lentiviral infection [253]. Additionally, BM-MSCs-derived exosomes have shown a similar therapeutic potential [134].

However, due to their proximity with the HSC and hematopoiesis processes, their impairment, in particular through a viral infection, could be critical for BM-related diseases. Moreover, given bone marrow engraftment is the unique suitable treatment for certain diseases (e.g., hematological malignancies), the persistence of virus-infected cells in grafts could represent a major risk, especially in patients that are usually immunocompromised.

BM-MSCs have been shown to be targeted by a variety of viruses during the natural history of a clinical infection.

Human Immunodeficiency Virus (HIV), that causes Acquired Immuno Deficiency Syndrome (AIDS), was shown to infect BM-MSC and integrate its DNA in BM-MSC’s genome [29,188]. Also, both intra- and extracellular interactions of HIV proteins Tat and Nef with BM-MSCs were observed [188,189]. This leads to an impaired osteoblastic/proadipogenic differentiation therefore promoting respectively decreased bone marrow density and fat toxicity described in HIV-infected patients [188,189]. Furthermore, HIV is able to induce senescence of BM-MSCs through its proteins Tat and Nef, but also p55-gag [27,189]. BM-MSC senescence has been associated to HIV infection-related cytopenia [27]. Finally, a role of BM-MSCs in HIV-related disease is the reactivation of HIV in latently-infected cells [187].

*Herpesviridae* is a large family that includes diverse viruses causing human diseases, such as Human Cytomegalovirus (HCMV), or Human Herpesviruses (HHV). Among these viruses, HCMV, HHV-1, HHV-3 (or Varicella-Zoster Virus, VZV) and HHV-8 (Kaposi’s Sarcoma-Associated Herpesvirus) exhibited the ability to infect BM-MSCs [30,190,192,193,254]. HCMV is a leading cause of congenital birth defects, as well as the major cause of diseases in immunocompromised individuals, notably following organ or BM transplant.

Of note, BM is an important site involved in the pathogenesis of chronic HCMV infection. The virus establishes latency in hematopoietic progenitors and can be transmitted after reactivation to neighboring cells. HCMV has deleterious effects on BM-MSCs function, changing the repertoire of cell surface markers expression (CD29, CD44, CD73, CD105, CD90, MHC class I and ICAM-1), and modifying the physiological interaction between BM-MSCs and HSC [191]. In addition, similarly to HIV, HCMV alters BM-MSCs biology and might contribute to the development of diseases (impairment of osteoblast regeneration, cartilage regeneration, hematopoiesis and properties/functions of immune progenitor cells), due to a deterioration of BM-MSC differentiation capabilities [191]. More strikingly, BM-MSCs were evoked as a potential reservoir for HCMV [30], which could be crucial for treatments involving BM engraftment.

HHV-8 also induced alterations of BM-MSCs, infected cells displaying lower proliferation rates and altered expression of Kaposi’s Sarcoma markers as well as altered phenotypes related to malignant transformations [192].

Even if BM-MSCs are physically less exposed to respiratory viruses, the influence of these viruses on BM-MSCs has been explored in several studies [194,195,196]. Avian Influenza A (H5N1) virus productively infects BM-MSCs provoking cell death and IL-6, CCL2 and CCL4 secretion by co-cultured monocytes [195]. The link between these findings and abnormal hematologic clinical descriptions such as lymphopenia, thrombocytopenia, and pancytopenia observed during avian flu remains to be addressed. Still, as for HCMV, the infection of BM-MSCs by Influenza virus might represent a risk of transmission during BM transplantation [194]. BM-MSCs contribute to Respiratory Syncytial virus-related lung disease. Indeed, this virus is able to infect and replicate in BM-MSCs, altering their immunoregulatory functions via an increase of IFN-β and IDO expression [196]. Zika virus (ZIKV) is a recently reemerging flavivirus, responsible for dengue-like syndrome in most of the cases but also associated with Guillain-Barré syndrome (GBS) in severe cases [230]. ZIKV, which is more known for its implication in neurological induced disorders (as described in *Brain pericytes* section), has the ability to infect and replicate in BM-MSCs [197]. BM-MSCs infection by ZIKV causes increased IL-6 expression and impaired osteoblast differentiation, pointed out by a decreased expression of alkaline phosphatase (ALP) and Runt-related transcription factor 2 (RUNX2) [197]. These data demonstrate the potential involvement of ZIKV in the development of bone pathologies. Finally, BM-MSCs might serve as an extrahepatic reservoir for Hepatitis B virus (HBV). Indeed, BM-MSCs are infected in vitro by HBV [198]. Moreover, BM-MSCs are able to transport HBV to injured tissues, as evidenced by transplantation of BM-MSCs in a mouse model of myocardial infarction, resulting in HBV infection of injured heart and other damaged tissues [198]. Thus BM-MSCs might play a critical role in HBV-associated myocarditis and other HBV-related extrahepatic diseases.

BM-MSCs are best known for their immunoregulatory activities during injury as aforementioned. They possess antiviral effector functions (Figure 3), even if the innate immune response of BM-MSCs against each virus previously cited has not been examined in great depth. BM-MSCs are basically expressing several cytosolic PRR albeit at low levels [10].

Yet in the context of a viral infection, it has been demonstrated (using Poly I:C to mimic RNA viruses) that BM-MSCs can up-regulate their PRR expression [255]. Among the up-regulated cytosolic PRR, Retinoic acid-inducible gene-I (RIG-I)-like receptors (RLR: RIG-I, Melanoma-Differentiation-Associated Gene-5 or MDA5), and Toll-Like Receptor 3 (TLR3) are important to detect viral RNA [255,256]. After sensing, BM-MSCs engage different cell signaling pathways according to the stimulated PRR. A TLR3-dependent sensing activate mitogen-activated protein kinase pathways (through p38 MAPK and p46 JNK) [256]. Whereas RLR-dependent sensing stimulates IFN signaling pathway through TBK1/IKK-ε and subsequent interferon regulatory factor (IRF) 7 phosphorylation [255]. These signaling pathways both trigger the production of pro-inflammatory cytokines and peptides with antiviral activities. Hence, in viral context BM-MSCs express increased levels of IL-1β, IL-6, IL-8, IL-11, IL-12p35, IL-23p19, IL-27p28, TNF-α, and CCL5/RANTES [255,256,257]. Furthermore, BM-MSCs produce IFN-β and IFN-λ1 in a RIG-I-dependent manner in contrast to other MSC subtypes producing it in both TLR3 and RIG-I dependent manner [255]. Classically, type I IFNs (such as IFN-β) induce the expression of Interferon Stimulated Genes (ISG) by the interaction with the IFN receptor (IFNAR). To date, the expression of ISGs has not been explored in BM-MSCs and needs further investigations. Interestingly, an antiviral activity of IDO has been demonstrated by decreasing HCMV and HSV-1 replication in BM-MSCs [258].

Of note, exosomes derived from allogenic BM-MSC (e.g., EXOFLO^TM^) have been used clinically to limit successfully the cytokine storm and the associated tissue injuries in patient with COVID-19 [259]. This would argue that MSC in tissues may be able to control the infection mediated by SARS-CoV-2.

#### 3.1.2. Lung Resident-MSC and Viruses

To date, lung resident-MSCs (LR-MSCs) have been mostly studied in the context of fibrosis and bronchiolitis obliterans syndrome, notably after lung transplantation [52,260]. Several important and recent gene tracing studies argued for a critical role of perivascular MSC, derived notably from NC progenitors, in lung diseases [143,147]. However, LR-MSCs have been poorly studied in the virology field even if their pivotal position makes them potentially susceptible to viruses and particularly respiratory viruses. Supporting this, possible interactions between SARS-CoV-2 and MSC have been envisaged in COVID-19 positive patients experiencing pericytes loss and apoptosis (observed by cleaved-caspase 3 immunostaining) [199]. Because pericytes often include MSCs, as discussed above, these findings suggest LR-MSCs as potential targets of SARS-CoV-2 in the lung [200]. Currently available data regarding the expression of ACE2 and TMPRSS2 in MSCs are discordant [261].

LR-MSCs have shown the highest permissiveness, replication rate and release during HCMV infection in cultured cells [30]. Moreover, LR-MSC is now recognized as a natural reservoir for HCMV after its detection in seven out of nine individual donors [30].

Lung PDGFβ+ MSC were recently shown as targets for HIV. These cells express both the primary receptor of HIV (CD4) and its major co-receptors (CXCR4 and CCR5), permitting productive infection and replication of HIV in vitro [201]. This finding may indicate a possible implication of LR-MSCs in the HIV-related pulmonary complications [201].

Surprisingly, in comparison to advances made for other MSCs subtypes, very few data are available to date regarding the specific antiviral response mounted by LR-MSCs during infection. Efforts should be made to fill this gap since it has already been demonstrated that a different antiviral response may be mobilized by tissue-specific MSCs [255].

#### 3.1.3. Adipose Stem Cells and Viruses

Adipose Stem Cells (ASCs) are a subtype of MSCs found in higher amounts than BM-MSCs and their isolation is easier, safer, less painful and less time consuming using liposuction techniques [50,262]. Consequently, they became an alternative to BM-MSCs for cell-based or exosome-based therapy using MSCs [263,264]. ASC can differentiate into different lineages and including into a Schwann-cell like phenotype expressing S100, GFAP with neurite outgrowth activity [265]. They are now studied to be used in a broad range of diseases [263]. MSCs are highly sensitive to their environment, influencing their responses through notably TLR or RLR [255]. ASCs have important immunomodulatory functions shown by their high cytokine response [255]. Notably, viral stimuli induce an important activation of these cells [255]. Furthermore, the importance of immunoregulatory function of ASC is highlighted by their capacity to produce various adipokines (adiponectin, leptin) that influence the inflammatory responses in most tissues [266]. Most of the studies regarding adipose tissue and viral infection have focused on adipocytes which are the main cell type of adipose tissue and also a derivative of ASCs. In the context of HIV, anti-viral treatments were shown to accumulate in adipocytes and alter their adipokine production, explaining some long-term complications of the therapies [267]. Adipocytes are mostly resistant to direct viral infection by HCV or HBV [268,269]. Due to their differentiation capabilities and their accessibility, ASC represents a good candidate for in vitro cell infection model [268,269], research on factors influencing viral replication like miR-27a in HCV infection [269] or stem-cell based therapies during HBV chronic infection [268]. Nevertheless, further studies are needed to explore the direct role of ASC in the immune response in the course of viral infection and addressing the regulation in adipokine and miRNA production.

#### 3.1.4. MSC of the Liver: Hepatic Stellate Cell/Ito Cell and Viruses

Hepatic Stellate Cells (HepSC, also named Ito cells) have been firstly described by Von Kupffer in the 19th century [70,270], for review, [13,18,271]. Using gene tracing experiments, they can be derived from MPZ+ NC cells and they express GFAP (astroglial marker) particularly in response to injuries (e.g., [12]). In physiological conditions, HepSCs are mostly long-lived quiescent cells residing in the perisinusoidal space of Disse [70,272]. They represent 15% of resident liver cells [273] and are fat-storing, vitamin-A rich, with presence of long processes [70,271,274]. These cells interact with neighboring cells like hepatocytes, resident macrophages (Kupffer cells), endothelial cells or nerves [70,273]. Also, they express markers of mesenchymal origin like desmin, alpha SMA [272,273]. Under pathological conditions, they show proliferative activity and differentiate in myofibroblast-like cells [70]. Activated HepSCs produce large amounts of ECM and matrix metalloproteases (MMPs) [70,273,275,276].

Quiescent HepSCs participate in physiological conditions to liver homeostasis, regeneration, development, retinoid metabolism, extracellular matrix homeostasis, and drug metabolism [277]. Upon liver injury, various stimuli, depending in the liver disease, activate HepSCs. The chronic activation of HepSC leads to excessive ECM accumulation and liver fibrosis [206,277].

In the context of viral infection, liver fibrosis represents a critical complication of end-stage liver disease progression. It mainly occurs through the persistent activation of HepSCs during chronic infection. This process has been studied during either HCV, HBV, or HIV pathologies.

HepSC activation occurs through direct or indirect viral effects [207]. HCV can directly activate HepSCs via either E2 protein binding to CD81 [208,277], NS3-NS5 proteins [209,277] or dsRNA [277,278]. HepSC expression of CCR5 and CXCR4 co-receptors for HIV has been described [202]. Moreover, HepSC infection by HIV has been established in vitro but not in vivo [202,203]. HIV through its envelope protein gp120 has been shown to induce HSC activation and chemotaxis [202].

The indirect HepSC activation also represents an important mechanism. In fact, the hepatic inflammatory environment is an important factor of HepSC activation thus leading to pro-fibrogenic factor release [277]. Hepatocytes produce profibrogenic factors like TGF-β1 or TIMP-1 [204,277], apoptotic bodies [277] or ubiquitin carboxy-terminal hydrolase L1 (UCHL1) [205] during HCV infection. Also, activated HepSC will release cytokine (i.e IL-1α) which will further stimulate hepatocytes to produce pro-inflammatory cytokines (IL-6, IL-8) [277]. In the course of HBV infection, lymphocytes are implicated in inflammatory tissue damages [210]. As shown for the hepatocytes, activated HepSCs may participate in the activation of immune cells thus amplifying the local inflammatory environment. For example during HBV infection, HepSCs could promote Th17 cell activation through IL-17RA-dependent proinflammatory cytokine expression [211]. Similarly, during HIV chronic infection, many factors participate in HepSC activation and fibrogenesis like NK cells dysfunction, intestinal microbial translocation, Kupffer cell inflammatory response, hepatocytes apoptosis, and liver damages [202].

The implication of HepSCs in the context of liver fibrosis during chronic viral diseases has been highly studied. However, their role during acute hepatotropic viral infection notably the importance of their immune response for viral clearance has been less explored.

#### 3.1.5. MSC of the Kidney: Mesangial Cell and Viruses

Mesangial cells (MCs) is the name used to designate the resident perivascular MSCs of the kidney and chiefly involved in fibrosis [4,144,147]. They are derived from the NC and responsible for mesangium matrix formation and glomerular capillaries support [12]. As for other MSCs, their location at the periphery of the vessels allows them to act as gatekeepers and to sense their environment in case of injury such as viral infection [14]. Similar to what is observed in other organs, viral infection in the kidney may lead either to a direct deleterious viral effect on targeted cells or to an indirect effect, due to an inappropriate immune response. MC participates in both mechanisms. Indeed, MCs were demonstrated to be targeted by various viruses [212,214,219,279,280,281,282,283]. First, HIV was shown to infect MCs [212] with an orphan G protein-coupled receptor 1 (GPR1) as a co-receptor [213]. Additionally, HIV affects MCs either directly or indirectly leading to proliferation and matrix synthesis, thus MCs play a role in HIV-associated glomerulosclerosis [28]. HCMV is the most threatening pathogen after a kidney transplantation, due to immunosuppression [284] and potential virus rebound in immunocompromised host patient. HCMV is associated with the development of glomerulosclerosis due to the matrix deposition caused by MCs [214,215]. Of note, MCs were shown to be targeted by HCMV and to allow its efficient replication [216,217], even if the link between this infection and the pathology associated is poorly understood [214]. Among the viruses of critical interest, HCV is a major problem worldwide, despite recent therapeutic improvements [285]. The pathology associated with HCV is mostly liver disease, but with frequent extrahepatic complications, such as glomerulonephritis. HCV triggers TLR3 activation of MC leading to the release of procoagulant factors that causes vascular thrombosis and finally glomerulonephritis [279]. Furthermore, MCs may be infected by ZIKV. It has been demonstrated that these cells may serve as a reservoir for the virus, thereby this finding may explain the high level persistent viruria observed [219]. However, to date, no link has been established between ZIKV infection of MCs and kidney disease.

Of note, the antiviral response of MCs was not studied in the context of the viral infection aforementioned. However, viral RNA and DNA trigger a common antiviral response in MCs notably through RIG-I, MDA5 and TLR3 [286,287,288]. This response includes notably Interferon-type I response and secretion of proinflammatory cytokines and chemokines.

On the other hand, MCs also participate in inappropriate immune responses in case of infection. Indeed, immune complexes associated with viral RNA or DNA triggers overwhelmed immune response leading to glomerulonephritis. This mechanism was notably shown in a mouse model of Immunoglobulin A nephropathy (IgAN) induced by Sendai virus [280], but also in patients suffering from Lupus nephritis [289].

#### 3.1.6. Brain MSC: Brain Pericytes and Viruses

Brain pericytes (BPs) are the resident MSCs of the brain [68]. BPs are essentially derived from NC [11,15]. With endothelial cells, and astrocytes, they constitute the blood-brain-barrier (BBB), the specialized vascular unit of the brain, responsible for its protection from external factors. Due to this major role in brain defense, an infection of BPs could lead to BBB disruption and increased permeability. BPs are well-equipped to ensure their sentinel function. They were shown to express PRR, notably TLR9, allowing them to be responsive in case of microbial infection, particularly by non-canonical inflammasome activation [290]. Moreover, BPs are immunocompetent cells with NO, IL-1β, IL-3, IL-9, IL-10, IL-12, IL-13, TNF-α, IFN-γ, G-CSF, GM-CSF, Eotaxin, CCL3, and CCL4 secretion following LPS stimulation [291]. Of note, BPs may also participate in the canonical antiviral response. Indeed, in response to IFN-γ and TNF-α, they showed upregulation of pro-IL-1β and pro-Caspase 1 mRNA expression [290]. Moreover, BPs express CCL2 after Poly:IC or LPS stimulation, relaying inflammatory signals from circulation to neurons, leading to elevated excitability [292].

Additionally, due to their location and the different route of entry for foreign agents (notably receptor-mediated endocytosis, unspecific transport by pinocytotic vesicles), BPs represent possible targets for viral infection. HIV infection can cause BBB disruption and contributes to the development of neurological dysfunction (e.g., HIV-associated neurological disorders—HAND). HIV is able to infect and replicate at low levels in BPs, using its co-receptors CXCR4 and CCR5 [222,293]. These cells even constitute one of HIV reservoirs in the central nervous system (CNS), the infection reactivation being possible thanks to genome integration mechanism. An increase in integrated genome in BPs is leading to a latent stage of infection following the initial peak of HIV production in the CNS [294,295]. Moreover, HIV infection of BPs has deleterious effects on BBB. Of note, it was shown that a decreased BPs coverage of BBB was a consequence of BPs dysfunction in HIV-infected patients [223], BPs coverage being negatively correlated with BBB permeability. The loss of pericytes observed has been associated with early PDGF-BB expression, which promotes pericytes migration away from their perivascular location [221]. Moreover, HIV infection leads to IL-6 secretion [222]. This may be another consequence of PDGF-BB secretion and downstream receptor signaling events [296]. IL-6 secretion by BPs also concurs in HIV-induced CNS damage and BBB disruption observed [222], the proinflammatory-induced response of endothelial cells causing BBB impairment [297].

HCMV infection may cause neurological pathologies either in children or immunocompromised patients. If, astrocytes and endothelial cells were initially reported as the main targets of HCMV at the BBB level [224], BPs were recently shown to be more permissive and sensitive to HCMV-induced lysis [225]. Thus, BPs are contributing to HCMV dissemination in CNS [30,225], as well as neuroinflammation via CXCL8, CXCL11, CCL5, TNF-α, IL-1β, and IL-6 secretion [225].

Japanese Encephalitis Virus (JEV) is a neurotropic mosquito-borne flavivirus that causes encephalitis, with neurological and psychological sequelae among a majority of survivors. Neuroinflammation is the most outstanding mechanism associated with JEV pathogenesis. BBB integrity is critical to regulate neuroinflammation, by limiting circulating immune cells entry and avoiding overwhelmed immune response. BPs are targeted by JEV as least in vitro. Following infection, they secrete IL-6 promoting proinflammatory responses and proteasomal degradation of Zonula Occludens-1 (ZO-1), thus participating in JEV-induced BBB impairment [226]. In addition, JEV was shown to trigger NF-κB through TLR7/MyD88 dependent axis, leading to PGE2 and CCL5/RANTES secretion [227]. The latter causes attraction and infiltration of leukocytes to the site of infection, provoking the aforementioned inappropriate immune response responsible for JEV neuropathology. The same mechanism is evoked for Herpes Simplex encephalitis but need to be further explored [228]. ZIKV, which is known for its neurological complications (e.g., microcephaly, Guillain-Barré Syndrome, and encephalitis) [230], can invade CNS through BPs infection at the level of the choroid plexus (via the receptor AXL) [229]. Both murine and human BPs are susceptible to ZIKV [229]. Furthermore, ZIKV-infected BPs are associated with BBB defect in vitro, illustrated by increased cytosolic ZO-1, decreased transepithelial electrical resistance and higher degree of FITC-dextran transport [229].

### 3.2. Osteoblasts and Viral Infections

Osteoblasts (OBs) are stromal cells, responsible for bone formation, through osteocalcin, ALP and type I collagen expression. As the other cells presented in this section, OBs (e.g., craniofacial bones) are derived from MSCs of the NC [298]. A great deal of pathogens are able to target bone tissues, leading to a variety of diseases ranging from caries to osteomyelitis. Among these pathogens, bacteria are the most cited. Yet the implication of viruses, and more specifically arthritogenic ones, in development of bone disorders should not be underestimated.

Ross River Virus (RRV) is a mosquito-borne alphavirus that generally causes flu-like illness and polyarthritis. As other alphaviruses, RRV might promote the development of bone diseases by targeting osteoblasts through Mxra8 [119]. Chen et al. addressed the effect of RRV infection on osteoblasts and showed that RRV-infected osteoblasts are producing high levels of IL-6 and CCL2 [233]. Additionally, this infection led to an imbalance in the Receptor Activator of Nuclear factor-KappaB Ligand (RANKL)/Osteoprotegerin (OPG) ratio in favor of osteoclastogenic activities and bone loss [233]. Moreover, the same team demonstrated an increased susceptibility to RRV in osteoblasts from osteoarthritic patients, further promoting the adverse effects of infection previously mentioned, due to a delayed IFN-β induction and RIG-I expression [232].

Similar outcomes are observed following OBs infection by Chikungunya virus (CHIKV), which is another arthritogenic alphavirus, best known for its recent epidemics between 2005 and 2006 in Reunion Island. Thus, an increased RANKL/OPG ratio in favor of osteoclastogenesis [234,299] can lead to bone loss through monocytic osteoclast recruitment [299].

HCV, besides its hepatic-related disease, might be able to trigger bone disorders. Indeed, HCV-infected patients have a higher risk of osteoporosis [300] and fracture [237]. Conversely, HCV infection was also associated with bone density hardening and osteosclerosis [235]. Once again, RANKL/OPG ratio impairment during HCV infection could be the cause of these detrimental consequences [235]. Kluger and colleagues addressed the permissiveness of OBs to HCV and provided evidence that OBs and osteoblast progenitors harbor HCV replication in the bone [236]. Thus, the implication of OBs in the pathophysiology of bone diseases following HCV infection should be explored given their involvement during alphaviruses-related disorders. On the other hand, as previously evoked, flavivirus ZIKV infection of BM-MSCs interferes with their ability to differentiate in osteoblasts, with a significant increase of IL-6 expression and a decrease of key osteoblast marker (e.g., ALP and RUNX2) in infected MSCs [197]. This susceptibility leads to impaired osteoblast function during ZIKV infection, triggering an imbalance in bone homeostasis and inducing bone-related disorders [197]. 

Measles morbillivirus (MeV), which belongs to the *Paramyxoviridae* family, is the cause of measles, a highly contagious disease. Symptoms of measles associate general symptoms (fever, cough) with a generalized maculopapular and erythematous rash. Moreover, MeV has been evoked in bone-related diseases such as otosclerosis [238,239] and Paget’s disease [240], both characterized by disturbed equilibrium of bone resorption and new bone formation. Several teams reported that productive OBs infection by MeV participates in the development of these disorders. Hence during MeV infection, OBs exhibit a higher expression of several osteogenic markers: bone morphogenic proteins (BMP-1, -4, -5, -6, and -7), ALP, bone sialo-protein, collagen 1α1 and OPG [238]. These findings highlight the ability of MeV to stimulate osteogenic differentiation of OBs thus participating in imbalance in bone formation. Such findings were corroborated by Potocka-Bakłażec et al., by demonstrating increased levels of TNF-α, IL-1β and decreased levels of OPG (conversely to Ayala-Peña et al. study) in MeV positive patients, also testifying bone remodeling subsequent to MeV infection [239].

HIV-infected patients have a higher incidence of osteopenia and osteoporosis due to bone demineralization and reduced bone mass. This is the result of OBs increased apoptosis following HIV infection in response both to expression of TNF-α and impaired Wnt/β-Catenin signaling in infected cells [231,301].

Osteoclasts, derived from macrophages, were thought to be the principal sensor of infection in the bone, due to their wide expression of PRRs. Notwithstanding, a growing body of evidence is in favor of PRRs’ expression by OBs, arguing for a sensing and antimicrobial role shared between osteoclasts and OBs [302]. Even if exacerbated inflammatory response in case of infection may lead to detrimental consequences, this has notably been exemplified earlier with IL-6 and development of arthritic pathologies, antiviral response remains essential to counteract most of virus-induced pathogenesis processes. Thus, the response of OBs in viral context has been addressed by Nakamura et al. on mouse osteoblastic cells (MC3T3-E1), using Poly:IC. They showed that in context of viral infection, OBs produce IFN-β as early as 1 h after stimulation due to Poly:IC recognition by TLR3, with a peak at 12 h [303]. Of note, they reported production of TLR3 and RIG-I in response to IFN-β, making OBs fully capable of viral dsRNA detection [303,304]. Of further note, IFN-β triggers CXCL10 production through a IFN-α/β receptor-STAT1 pathway [303]. Taken together these data suggest the ability of OBs to mount a proper IFN-type I response. Additionally, IL-27, a cytokine regulating immune responses as well as hematopoiesis and bone remodeling, was found to be expressed by OBs in inflammatory conditions (e.g., presence of type I IFN, IL-1β and TNF-α) [305]. So, OBs are thought to be part of a negative-feedback mechanism, limiting bone erosion and dampening T cell-mediated immune pathology during bone inflammation (therefore antiviral response).

### 3.3. Schwann Cell of Peripheral Nerves and Viral Infection

Schwann Cells (SCs) originate from NC. Moreover, MSCs are able to give rise to Schwann Cell-like cells after induction [306,307], these cells being able to drive a proper myelination. SCs are support cells that have a pivotal role in myelination of neurons from the peripheral nervous system (PNS). Thus, their affection is linked to demyelinating disorders of PNS (e.g., Guillain-Barré Syndrome).

As previously seen, HIV can cause neurological disorders, called HIV-associated neurological disorders (HAND). Among these HAND, we can notably cite distal neuropathy. Even if SCs are not primarily infected by HIV (infection has not been described in vitro), they express chemokine receptor CXCR4, a receptor for HIV-1 gp120 [241]. The pathway driven by HIV-1 gp120 leads to RANTES and TNF-α secretion, stimulating axon and neuron to release TNF receptor-1 (TNFR1) [241]. This promotes dorsal root ganglion neurotoxicity, including axon and myelin injury [241].

CNS tropism of Herpes Simplex Virus (HSV) and Herpes Zoster Virus/Varicella Zoster Virus (VZV) is well-established. Furthermore, infection of SCs by HSV or VZV was experimentally demonstrated [308,309]. However, it is difficult to say if this infection is of clinical relevance, since the principal mechanism evoked for HSV-induced GBS is a molecular mimicry of viral proteins, leading to cross-reacting antibodies [242]. However, this statement is based on case reports and needs to be further explored experimentally, in order to exclude any direct involvement of SCs in HSV-induced GBS.

Similarly, HCMV inclusions can be observed in SCs [243]. HCMV infection is the second most frequent infectious etiology of GBS [244], with notable cases in immunocompromised patients [245]. The pathogenesis of CMV-associated GBS have been linked to a probable molecular mimicry generating autoantibodies against, among others, moesin expressed by SCs [246]. Of note, these findings are called into question and need to be experimentally confirmed with an animal model, since no other team has obtained similar results [247].

As aforementioned, ZIKV is increasingly implicated in neurological disorders affecting CNS (i.e., microcephaly) as PNS (i.e., GBS) and of critical concern in the last years [230]. Initially, it has been demonstrated that CNS cells and oligodendrocytes, responsible for myelination in CNS, were more susceptible to ZIKV than PNS cells, causing axon and myelin injuries [248]. Yet more recent data indicate SCs are susceptible to ZIKV [310,311]. Volpi et al. showed that, in myelinating dorsal root ganglion explants from *Ifnar1^−/−^* mice, ZIKV infection of SCs leads to endoplasmic reticulum stress pathway activation and apoptosis in these cells, which finally cause demyelination and axon degeneration [310]. Additionally, Dhiman et al. showed a sustained viral production and a significant cell death at 96 h post-infection in vitro in human SCs, the infection inducing expression of proinflammatory cytokines (IL-6, TNF-α, IFN-β and IL-29) [311]. These works pave the way for a direct viral pathogenic effect or a cell-mediated inflammation in pathogenesis of ZIKV-associated GBS, even if a possible antibody dependent enhancement with DENV sera or a demyelination induced cross-reactivity of anti-ZIKV antibodies are also discussed elsewhere [312,313].

Primary antiviral response mounted by SCs has not been well studied for each virus previously presented. However, SCs possess an efficient immune system to detect pathogens, represented by several PRRs. Among them, TLR3 and TLR7 are PRR devoted to virus detection and expressed by SCs [314]. TLR3 and TLR7 trigger a classical antiviral response after PAMP recognition. This response implies IFN response, driven by NF-κB and IRF, ISG release, and ultimately apoptosis, subsequently to inflammatory cytokine stimulation of extrinsic pathway.

## 4. Concluding Remarks

MSCs have been mainly studied with the goal to mobilize their therapeutic activities to treat a large variety of acute and chronic inflammatory diseases and with the hope that they will not be hosting pathogens such as viruses. MSC-based immuno-therapies (and including EV-derived from MSC) may also involve lentiviral or adenoviral infections of MSCs to express recombinant factors to promote tissue repair while controlling fibrosis/calcification. Nardi and colleagues proposed in 2006 that MSC should also be considered as packaging cells for in situ gene therapy (or to deliver oncolytic viruses) of solid tumors given the tropism of MSCs for tumor microenvironment [315]. These experimentally-infected MSCs may produce viral vectors for long periods of time. In this review we are raising the awareness that MSCs are also strategically located to detect incoming pathogens such as viruses and to set a relevant immune response. It should be noted that the hormone procalcitonin (PCT) produced by MSCs is one of the best and earliest markers of a plethora of infectious diseases. The level of PCT is markedly increased long before CRP and is used particularly in intensive care units. The role of MSCs in innate immune sensing has been neglected, and yet, their unique perivascular location makes them ideal gatekeepers as well as targets for viruses and facilitating tissue invasion.

Further clinical and fundamental studies are now warranted to test for the capacity of viruses to use MSCs as a sanctuary and with long term effects. MSCs infection could contribute to many chronic pathologies as described herein, but also questioning MSCs transplantation safety issues. Growing evidence suggests an underestimated role of MSCs during viral infection, notably through the antiviral response mobilized by MSCs, adding a further skill to their immune functions. Our understanding on this subject is still in its infancy and future studies addressing MSCs susceptibility, immunity and stem/stromal cell behavior in the context of viral infection are now highly warranted.

## Figures and Tables

**Figure 1 ijms-23-08038-f001:**
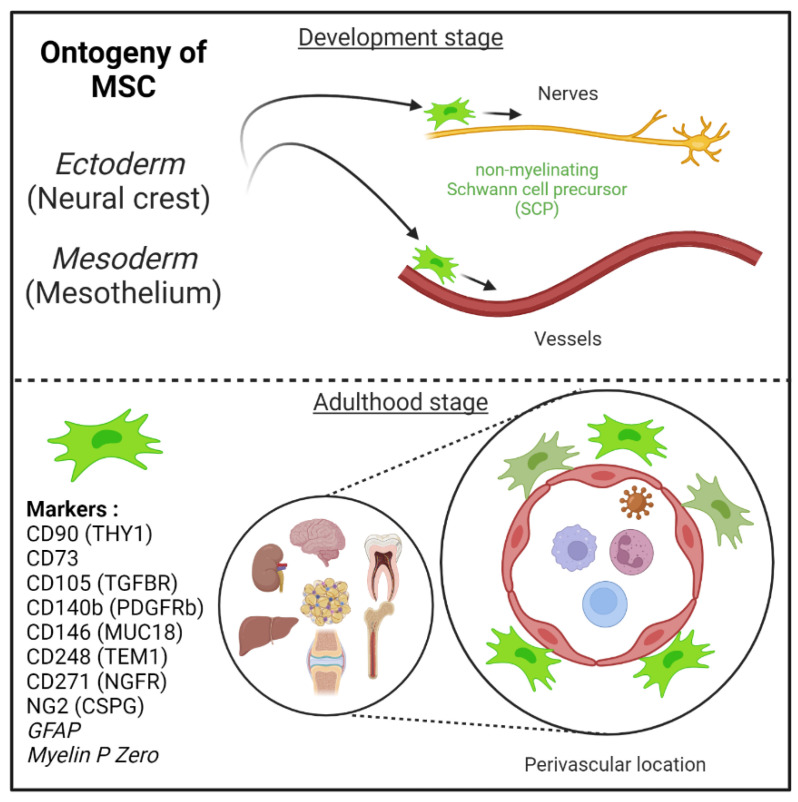
Mesenchymal stem cells (MSC) are derived from either he embryonic ectoderm (neural crest) or the mesoderm. MSCs can migrate along nerves and vessels during development and reside in virtually all post-natal organs and tissues. Along the nerves, MSCs are also known as non-myelinating precursor Schwann cells. Their location around vessels to form perivascular immune privileged niches has been demonstrated by several teams. MSC express several canonical markers which are differentially expressed in all major organs. CD271, GFAP, and MPZ are canonical neuroglial markers.

**Figure 2 ijms-23-08038-f002:**
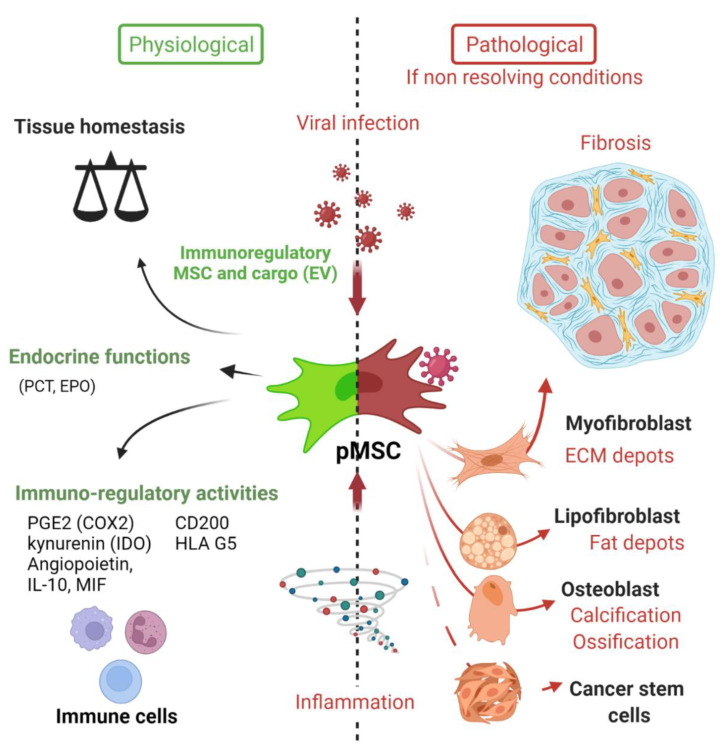
Deciphering the role of Mesenchymal stem cells in the context of viral infection. In physiological conditions, MSC have important immune functions to control viral infection as glatekeepers around vessels (perivascular MSC) and capable of mounting an innate immune antiviral response. MSC will also promote release of cytokines and chemokines to recruit immune cells to clear pathogens. Equally important is the expression of many immune regulatory factors to terminate the adaptive immune response to limit further cell injuries and promote tissue repair. Many viruses may infect directly MSC in tissues, thus remaining in an immunoprivileged niche favoring virus persistence, spreading and possible virus rebound in immunocompromised patients. Viruses associated to chronic inflammation (non-resolving) may also affect MSC differentiation (e.g., into myofibroblast) leading to excess of extracellular cell matrix production and contributing to organ dysfunction. Importantly, allogenic MSC and derived extracellular vesicles (EV) are nowadays important immunoregulatory cargo injected to patients for the treatment of inflammatory-infectious diseases such as COVID-19. Safety issues are nevertheless highly warranted. PCT (pro-calcitonin); EPO (erythropoietin) hormones.

**Figure 3 ijms-23-08038-f003:**
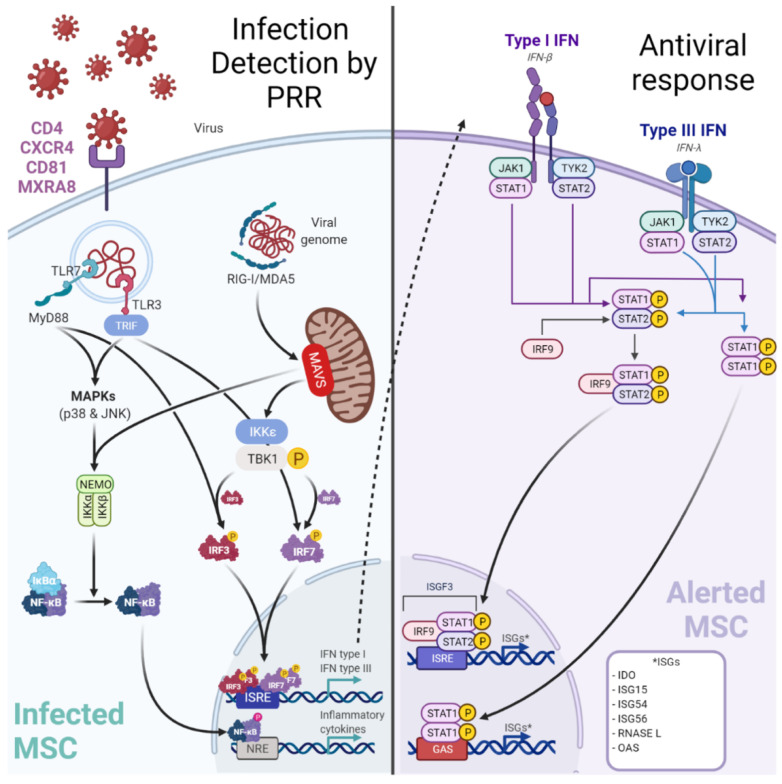
Infection of MSC by viruses may be controlled by a canonical innate immune response. Viruses may target perivascular MSC naturally expressing receptors (e.g., MXRA8/alphavirus) to grant entry. Among the up-regulated cytosolic pattern recognition receptors (PRRs), Retinoic acid-inducible gene-I (RIG-I)-like receptors (RLR: RIG-I, Melanoma-Differentiation-Associated Gene-5 or MDA5) and Toll-Like Receptor 3 (TLR3) are important to detect viral RNA. After sensing, MSCs engage different cell signaling pathways according to the stimulated PRR. A TLR3-dependent sensing activates mitogen-activated protein kinase pathways (through p38 MAPK and p46 JNK). RLR-dependent sensing stimulates IFN signaling pathway through TBK1/IKK-ε and subsequent interferon regulatory factor (IRF) 7 phosphorylation. These signaling pathways both trigger the production of pro-inflammatory cytokines and peptides with antiviral activities. Hence, in viral context MSCs express increased levels of IL-1β, IL-6, IL-8, IL-11, IL-12p35, IL-23p19, IL-27p28, TNF-α and CCL5/RANTES to recruit and activate adaptive immune cells (T/B lymphocytes). Furthermore, MSCs produce IFN-β and IFN-λ1). Classically, type I IFNs (such as IFN-β) induce the expression of Interferon Stimulated Genes (ISGs, e.g., RNASEL) by the interaction with the IFN receptor (IFNAR).

**Table 1 ijms-23-08038-t001:** The different names given to Multipotent Mesenchymal Stem/Stromal Cells (MSC) and their tissue localization.

Tissue	Human	Reference
Bone Marrow	Bone Marrow-Mesenchymal Stem Cell *or* Adventitial Reticular Cell *or* Myoid Cell	[17,63,64,65]
Adipose tissue	Adipose Stem Cell	[50,51]
Intervertebral disc	Skeletal Progenitor Cell	[66]
Synovial membrane	Synovial Membrane-derived MSCs Fibroblast-like synoviocytes (FLS)	[53,67]
Amniotic fluid	Amniotic Stem Cell	[62]
Cord blood Placenta	Umbilical Cord Blood Stem Cell *or* Wharton’s Jelly derived MSC	[60,61]
Dental tissue	Dental Pulp Stem Cell *or* Stem cells from Human Exfoliated Deciduous teeth *or* Periodontal Ligament MSC *or* Stem Cell from Apical Papilla *or* Dental Follicle Precursor Cell	[55,56,57,58,59]
Kidney	Mesangial Cell	[4]
Brain	Pericytes, perivascular fibroblasts	[68,69]
Liver	Hepatic Stellate Cell (HSC) *or* Perisinusoidal Cell *or* Ito Cell	[54,70]
Lung	Human Bronchial Fibroblasts *or* Lung-resident MSC	[52]

**Table 2 ijms-23-08038-t002:** Genetic lineage tracing and single-RNA sequencing studies to identify mesenchymal stem/stromal cells in all major organs in health and diseases. Adapted from [94,152].

Tissue	Cell Markers	Lineage Tracing Marker	Location in Tissue	Model of Injury	Pathological Role	Reference
**Skin**	ADAM12+ PDGFRα+ CD140a + Sca1+	Adam12	Perivascular	Injury (Adjuvant, CFA)	Myofib.	[153]
	PDGFRα+ Dlk1+ Sca1/Ly6a+	PDGFRα, Dlk1	Low dermis	Wound	Myofib/Adipo.	[154]
	CD26/DPP4+	Engrailed-1	Tissue-resident	Wound	Myofib.	[155]
	Adiponectin+	Adiponectin	Intradermal adipocyte	Bleomycin-induced skin fibrosis	Myofib.	[156,157]
	Gli1+, PDGFRβ+, PDGFRα+ Nestin+	Gli1	Perivascular	-	Myofib.	[147]
**Skeletal muscle**	PDGFRα+, Sca1+, CD34+	-	Perivascular	Injury (notexin, Glyc)	Myofib/adipo.	[158,159]
	ADAM12+, PDGFRα+, Sca1+	Adam12	Perivascular	Injury (CX)	Myofib.	[153]
	Gli1+, PDGFRβ+, PDGFRα+	Gli1	Perivascular	I/R injury	Myofib.	[147]
	PDGFRα+, Sca1+ Clusterin+ Osr1+ progenitors	Prrx1	-	Barium Chl., Glycerol, Sciatic nerve injuries	Adipo/Osteob.	[160]
**Heart**	Gli1+, PDGFRβ+, PDGFRα+	Gli1	Perivascular	Angiotensin 2-fibrosis	Myofib.	[147]
	Tcf21 + Vimentin+, PDGFRα+, α-SMA+	Periostin	Tissue-resident	I/R myocardial fibrosis	Myofib.	[161]
**Kidney**	PDGFRβ+, CD73+	FoxD1	Perivascular, peritubular	UUO, I/R injury	Myofib.	[143]
	PDGFRβ+, CD73+ CD271(P75) + EPO+	Myelin P Zero	Perivascular	UUO/Folic acid injury	Myofib.	[12]
		Alpha SMA	Interstitial Fib	UUO	Myofib.	[162]
	Gli1+, PDGFRβ+, PDGFRα+	Gli1	Perivascular	UUO, I/R injury	Myofib.	[147]
	POSTN+ PDGFRβ+	PDGFRβ		Mo/Human	Myofib	[163]
**Liver**	PDGFRβ+, desmin+	LRAt Col1α1	Perivascular	CCl4-, TAA- injuries	Myofib.	[148]
	Vit A+ GFAP+, CD90- Desmin+ Calcitonin a	Collα1	Perivascular HSC and portal Fib	CCl4-, bile duct ligation injuries	Myofib.	[164]
	Gli1+, PDGFRβ+, PDGFRα+	Gli1	Perivascular	CCl4	Myofib.	[147]
**Pancreas**	2 subsets: PDGFRβ+ PLP1, and PMP22 (Schwann)	-	Stellate cells	Carcinoma	Collagen+ Stellate cells Neural crest Schwann-like, Cancer	[165,166]
**Lung**	PDGFRβ+, NG2+	NG2	Perivascular	Bleomycin injury	Myofib.	[167]
	PDGFRβ+, NG2+	FoxD1	Perivascular tissue-resident	Bleomycin injury	Myofib.	[168]
	Gli1+, PDGFRβ+, PDGFRα+	Gli1	Perivascular	Bleomycin injury	Myofib.	[147]
	Axin2+, PDGFRα–	Axin2 PDGFRα Wnt2	Perivascular Peribronchial	Bleomycin, naphthalene injuries	Myofib.	[169]
	Acta2-	Acta2 Adrp	Perivascular	Bleomycin injury	Myofib. and reversion to lipofib.	[170]
	Collagen+	Collα1	Perivascular and interstitial	Bleomycin injury	Fib. Clusters (11) of collagen-producing subpopulations including highly pathogenic cluster 8: cthrc1+	[171]
**Spinal cord**	PDGFR a/β+, CD13+ desmin–	Glast	Type-A pericytes	Spinal cord injury	Myofib Scar tissue	[142]
**Brain** **cortex**	PDGFR β+	Glast	Type-A pericytes	Cortico-striatal stab wound, EAE autoimmunity Mid. cer. Art.occlusion (MCAO)	Myofib Scar tissue	[172,173]
	-	Collα1	Type-A pericytes	EAE autoimmunity	Myofib. and Immune stroma	[174]
	-		perivascular	Healthy CNS	Three cluster of perivascular fibroblasts including type I fibulin+ (ECM^high^)	[69]
**Bone Marrow**	CXCL12+	Nestin	Pericyte-like	-	Stromal cells of HSC	[64]
	PDGFR a+ CXCL12/SDF1a+	SCF LepR	Pericyte-like	-	Myofibroblast, osteocyte, chondrocyte, adipocyte	[175,176,177]
	Gli1+, PDGFRβ+, PDGFRα+ Nestin+	Gli1	Perivascular	-	Myofib.	[147]
	CD105+ (TGFbR) Nestin+ CD73+ CD44+, CD29+, Sca1+, and CD51+	Gli1	Perivascular endosteal niche	Thrombo- poietin-induced fibrosis	Myofibroblast	[151]
**Synovial tissue**	CD90/THY1+ FAP+ CD248+	-	Sublining Fib.	Rheumatoid Arthritis (RA)	Immune effector profile: high expression of cytokines and chemokines	[163,178,179,180,181,182]
	CD55/DAF+ CD90- FAP+ PPDN+	-	Lining Fib.	RA	Bone effector profile: mediate bone and cartilage damage	[163,178,179,180,181,182]

**ADAM12**: a disintegrin and metalloprotease, originally named meltrin α, a protein involved in muscle cell fusion. Two naturally occurring human ADAM12 are known ADAM12-Long and ADAM12-Short. **DLK1**: Delta-like 1 homolog, also known as preadipocyte factor 1 (Pref-1) and Fetal antigen (FA1) is an EGF-like membrane-bound protein. It contains six EGF-like repeats followed by a region with a TACE (ADAM17)-cleavage site, a transmembrane domain, and a short intracellular tail. It’s a non-canonical NOTCH ligand. **EN1**: Engrailed 1, a homeodomain-containing transcription factor with essential, widespread roles in embryonic development. The expression of EN1 is silence after lineage commitment and not expressed under homeostatic conditions in adulthood. In response to TGF-β1, EN1-positive cells give rise to a subpopulation of fibroblasts with high ECM production and important roles in wound healing in adult murine skin. **Adiponectin**: a cytokine secreted by adipocytes, known for regulating glucose levels, lipid metabolism, and insulin sensitivity through its anti-inflammatory, anti-fibrotic, and antioxidant effects. **PRRX1**: Paired related homeobox 1, a master transcription factor of stromal fibroblasts for myofibroblastic lineage progression. **Osr1**: Odd skipped-related 1 transcription factor marks a subset of embryonic cells that constitute a developmental fibro-adipogenic precursor (FAP)-like population, which supports embryonic myogenesis also a developmental source of adult muscle interstitial FAPs. **Tcf21**: Transcription factor 21 (TCF21), also known as pod-1, capsuling, or epicardin. Expressed by heart fibroblasts even in resting cells in adult. **EPO**: Erythropoietin hormone produced by the kidney, and is mostly well-known for its physiological function in regulating red blood cell production in the bone marrow. EPO has additional organ protective effects. **GLI1**: amplified gene named for the GLIoma tumor in which it was discovered. It is a transcription factor downstream of the transmembrane smoothened (SMO) and activated after the binding of Sonic Hedgehog (shh). **CTHRC1**: Collagen triple helix repeat containing-1 and has been identified as cancer-related protein. **Periostin (POSTN)**: cell adhesion protein initially described from a mouse osteoblastic cell line. It contains vitamin K-dependent γ-carboxyglutamic acid (Gla) residues which are found in a small group of proteins called Gla-containing proteins. **PLP1**: Proteolipid protein 1 and spliced isoform, DM20, major components of myelin proteins in the CNS and peripheral nerves (Schwann cells). **PMP22**: Peripheral myelin 22, a tetraspan glycoprotein mainly expressed in myelinating Schwann cells and in the compact peripheral myelin. **Forkhead box D1 (FOXD1)**: was first identified in the forebrain neuroepithelium and has been demonstrated to be important in the development of kidneys and retina. **Myelin P Zero**: MPZ (P0), homophilic adhesion molecule of the immunoglobulin superfamily expressed by neural-crest-derived Schwann cell (peripheral nerve) and precursor Schwann cells. **Alpha SMA**: smooth muscle cell actin protein coded by ACTA2 gene. It is a canonical marker of differentiation of fibroblasts into myofibroblasts in response to TGF-β1. **LRAt**: Lecithin Retinol Acyltransferase localizes to the endoplasmic reticulum, catalyzes the esterification of all-trans-retinol into all-trans-retinyl ester. This reaction is important in vitamin A metabolism. **Collagen 1a**: Major protein of the ECM. **NG2**: Neuron-glial antigen 2 (known as chondroitin sulphate proteoglycan 4 (CSPG4), is a surface type I transmembrane protein involved in cell survival, migration and angiogenesis. **Axin2**: Axin-related protein with a role in the regulation of the stability of beta-catenin in the Wnt signaling pathway, like its rodent homologs, mouse conductin/rat axil. **ADRP**: adipose differentiation-related protein, also known as perilipin 2 (Plin2) or adipophilin, is protein involved in lipid droplet formation in the liver and peripheral tissues. **GLAST**: Glutamate/aspartate transporter it removes glutamate, major excitatory neurotransmitter, from the extracellular space. **LepR**: leptin receptor, also called obesity receptor (ObR), molecule that receives and transmits signals from leptin, a hormone released from adipocytes. **SCF**: Stem cell factor binds to c-Kit receptor and stimulates growth of hematopoietic stem/progenitor cells (HSCs) directly and/or by stimulating other cytokines. **FAP**: Fibroblast activation protein (FAP) with high expression in tumor stroma. It is a serine protease with dipeptidyl peptidase and endopeptidase activities, cleaving substrates at a post-proline bond. **PPDN**: Podoplanin, expressed by podocytes but also by cancer-associated fibroblasts (CAF). **CD55/DAF**: Decay-accelerating factor, GPI-anchored regulator of the complement system. **CD248**: Endosialin, tumor endothelial marker-1 highly expressed by perivascular cells and involved in carcinogenesis.

**Table 3 ijms-23-08038-t003:** Examples of viruses targeting MSCs and related pathologies.

Cells	Viruses	Related Outcomes
Bone marrow-MSC	HIV	Inability to support Hematopoietic Stem Cells expansion and implication in HIV-related cytopenia [27]; HIV-related reactivation [187,188,189]
	HCMV	Transmit to neighboring cells after reactivation [190,191]
		Modifies the physiological interaction between BM-MSCs and HSC
		Impairment of osteoblast regeneration, cartilage regeneration, hematopoiesis and properties of immune progenitor cells [191]
	HHV	Lower proliferation rates and altered phenotypes related to malignant transformation [190,192,193]
	Influenza A H5N1	Risk of transmission during bone marrow transplantation [194,195]
	RSV	Alteration of immunoregulatory functions [196]
	ZIKV	Impaired osteoblast differentiation and possible implication in development of bone pathologies [197]
	HBV	HBV-associated myocarditis and other HBV-related extrahepatic diseases [198]
Lung resident MSC/pericytes	SARS-CoV-2	Pericyte apoptosis and loss in COVID-19 patients [199,200]
	SIV/HIV	Development of HIV-related pulmonary complications [197,201]
Hepatic Stellate Cell (HepSC, Ito cells)	HIV	HepSC activation and chemotaxis through HIV gp120 envelope protein [202,203]
	HCV	HCV proteins as well as RNA released by hepatocytes are activating HepSC [204,205]. Constant activation leading to liver fibrosis [206,207,208,209]
	HBV	Release of IL-17 by infected cells which stimulate liver fibrosis by activation of HepSC [210,211].
Mesangial Cell (Kidney)	HIV	HIV-associated glomerulosclerosis due to increased proliferation and matrix synthesis [212,213]
	HCMV	Glomerulosclerosis [214,215,216,217]
	HCV	Glomerulonephritis [218]
	ZIKV	Viral reservoir (persistent viruria) [219]
	SARS-CoV-2	Stromal (MSC-like cells) are infected and may contribute to kidney fibrosis in a model of spheroid cultures and to be correlated with kidney fibrosis in COVID19 patients [220].
Brain Pericytes (BP)	HIV	Decreased BPs coverage of blood brain barrier (BBB) associated with higher permeability [221,222]
		IL-6 and PDGF-BB secretions concur in HIV-induced CNS damage and BBB disruption [221,223]
	HCMV	Contribute to HCMV dissemination [224]
		CXCL8, CXCL11, CCL5, TNF-α, IL-1β and IL-6 secretion causing neuroinflammation [225]
	JEV	IL-6 secretion leads to ZO-1 degradation and BBB impairment [226]
		PGE2 and RANTES secretion by BPs recruit leukocytes to the site of infection. Associated with BBB impairment, this provoke leukocyte infiltration and major neuroinflammation [227]
	HSV	BBB impairment associated with leukocytes recruitment leading to major neuroinflammation [228]
	ZIKV	Brain abnormalities and BBB defect [229,230]
Osteoblasts	HIV (gp120)	TNF-α and impaired Wnt/β-Catenin signaling promote bone demineralization and reduced bone mass leading to osteopenia and osteoprosis [231]
	RRV	Imbalance in RANKL/OPG ratio in favor of osteoclastogenic activities and bone loss [232,233]
	CHIKV	Proinflammatory (IL-6) and pro-osteoclastic (RANKL) effects in infected cells [234]
	HCV	Associated with bone density hardening and osteosclerosis [235]
		Increased risk of osteoporosis [236]
		Increased risk of fracture [237]
		Impairment of RANKL/OPG ratio [235]
	ZIKV	Impaired osteoblasts function triggering an imbalance in bone homeostasis and inducing bone-related disorders [197]
	MeV	Higher expression of several osteogenic markers and osteogenic differentiation [238]
		Otosclerosis [239]
		Paget’s disease [240]
Schwann Cell (SC)	HIV	Dorsal root ganglion neurotoxicity, including axon and myelin injury [241]
HSV/VZV	The principal mechanism evoked for HSV-induced GBS is a molecular mimicry of viral proteins [242]	
	HCMV	Probable molecular mimicry generating autoantibodies against moesin expressed by SCs [243,244,245,246,247]
	ZIKV	Possible direct viral pathogenic effect or a cell-mediated inflammation in pathogenesis of ZIKV-associated GBS [248]

## Data Availability

Not applicable.

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
