# Peer review of "Perivascular Mesenchymal Stem/Stromal Cells, an Immune Privileged Niche for Viruses?"

_ijms, 2022, doi:10.3390/ijms23148038_

Round 1

Reviewer 1 Report

The review by Lebeau et al. titled "Perivascular mesenchymal stem/stromal cells, and immune privileged niche for viruses?" is an interesting review of this fascinating topic. The review is relatively well written with some editing still required. The review is very comprehensive and summarizes the field relatively well. However, the review would benefit from addressing the following points:

1) Make sure to highlight the purpose of the review in the abstract, introduction, conclusions, and direct the writing to that purpose.

2) What are the authors' conclusions on the topic?

3) Are MSCs a significant carrier of viruses?

4) Does MSC virus carrying have an in vivo effect that is significant?

Author Response

Reviewer 1: The review by Lebeau et al. titled "Perivascular mesenchymal stem/stromal cells, and immune privileged niche for viruses?" is an interesting review of this fascinating topic. The review is relatively well written with some editing still required. The review is very comprehensive and summarizes the field relatively well. However, the review would benefit from addressing the following points:1) Make sure to highlight the purpose of the review in the abstract, introduction, conclusions, and direct the writing to that purpose.2) What are the authors' conclusions on the topic?3) Are MSCs a significant carrier of viruses?4) Does MSC virus carrying have an in vivo effect that is significant?

We thank the reviewer. The manuscript has been greatly modified to respond to the different questions raised by the reviewer. This is particularly and comprehensively addressed in the table 3 listing every single tissue-type MSC and the different viruses known to infect and have an impact on MSC’s critical functions. This is indeed the case for BM-derived MSC whereby viral infection will cause impairment at the level of the HSC stem cell niche and leading to anemia and immunosuppression (immune cell cytopenia). HCMV can reside in MSC, protected from the immune surveillance and will rebound on the ground that the patient may receive immunosuppressive drugs for example post-graft. MSC may not be directly infected by viruses but may be polarized by cytokines produced by neighboring epithelial-infected cells (e.g. hepatocytes infected by HBV releasing high levels of IL-17). At the level of an organ, MSC infection could lead to detrimental organ malfunctions as shown when viruses infect brain pericytes and affecting the blood-brain-barrier.

Reviewer 2 Report

Dear Editor,

This is a comprehensive review by Grégorie Lebeau et al that summarizes the scientific findings implicating a potential role for mesenchymal stem/stromal cells (MSCs) in serving as reservoirs for viral persistence. The review should be very useful to researchers investigating the immunomodulatory roles of MSCs in the pathobiology of chronic inflammation and fibrosis. While the scientific content is well described, the reviewer suggests the replacement of terms such as ‘magic bullet’ to make the content scientifically accurate. Additionally, the reviewer recommends proof-reading by a native speaker of the English language to correct minor grammatical errors in the manuscript text.

Author Response

Reviewer 2:

Dear Editor,

This is a comprehensive review by Grégorie Lebeau et al that summarizes the scientific findings implicating a potential role for mesenchymal stem/stromal cells (MSCs) in serving as reservoirs for viral persistence. The review should be very useful to researchers investigating the immunomodulatory roles of MSCs in the pathobiology of chronic inflammation and fibrosis. While the scientific content is well described, the reviewer suggests the replacement of terms such as ‘magic bullet’ to make the content scientifically accurate. Additionally, the reviewer recommends proof-reading by a native speaker of the English language to correct minor grammatical errors in the manuscript text.

We greatly appreciated the reviewer’s comments. The text has been corrected and the words ‘magic bullet’ have been removed.

Reviewer 3 Report

This review article aims to up-date the cell types in the family of mesenchymal stem cells, their functions and interactions with different viruses. However, it fails to up-date the recent progress and to clarify the field. Few valuable papers are reviewed, the most relevant being doi:10.1016/j.stem.2017.07.011. However, the recent use of single-cell RNA sequencing and signaling lineage reporters to generate a spatial and transcriptional map of the mesenchyme is not considered. Several recent studies found that each mesenchymal lineage has a distinct spatial address and transcriptional profile leading to unique niche regulatory functions. These studies define the cellular and molecular framework of mesenchymal niches and reveal the functional importance of developmental pathways in promoting self-renewal versus a pathological response to tissue injury. On this line, reviewing the viral infections and the mechanisms related to the mesenchymal cells would be of interest. 

Author Response

Reviewer 3:

This review article aims to up-date the cell types in the family of mesenchymal stem cells, their functions and interactions with different viruses. However, it fails to up-date the recent progress and to clarify the field. Few valuable papers are reviewed, the most relevant being doi:10.1016/j.stem.2017.07.011. However, the recent use of single-cell RNA sequencing and signaling lineage reporters to generate a spatial and transcriptional map of the mesenchyme is not considered. Several recent studies found that each mesenchymal lineage has a distinct spatial address and transcriptional profile leading to unique niche regulatory functions. These studies define the cellular and molecular framework of mesenchymal niches and reveal the functional importance of developmental pathways in promoting self-renewal versus a pathological response to tissue injury. On this line, reviewing the viral infections and the mechanisms related to the mesenchymal cells would be of interest. 

We thank the reviewer and have fully integrated his/her suggestions (see new table 2 including over 50 new references from 2010-May 2022). We are now providing a comprehensive review of the literature regarding the gene lineage tracing and single-cell RNA sequencing of the different MSC subpopulations in several major organs. To our knowledge, there is no data regarding the capacity of viruses to impact differently the different subsets of MSC and experiments along these lines are now highly warranted. Moreover, it remains to be established whether viruses can affect the differentiation potential of MSC either through transdifferentiation and/or differentiation towards a more pathogenic phenotype (e.g. MSC differentiation into myofibroblasts involved in fibrosis or osteoblast-like cells involved in vessel calcification).

Round 2

Reviewer 3 Report

The revised version of the MS is substantally improved. It still needs a clarification regarding the origin of MSCs (as pointed in page 1 line 46), to be also included in fig 1. 

Author Response

Reply N°II to reviewer 3:

We have fully responded to your criticisms and providing an additional set of information regarding the ontogeny of perivascular MSC on the ground that they are derived from either mesoderm (mesothelium) or/and ectoderm (Neural crest). This is now included in the revision N°II of our manuscript (within the text as presented below).

Figure 1 has been amended

Pr P. Gasque. 16/07/22

Origins of pericytes/perivascular MSC-fibroblasts derived from the neural crest and/or mesoderm embryonic tissues.

            The ontogeny of MSC before they rich their final position in adult tissues is still a matter of debate [1]. The identification of MSC relies on the characterization of genetic and protein markers (e.g. tyrosine kinase PDGF a or b receptors, Schwann cell myelin P zero, glioma-associated transcription factor Gli1, collagen, Acta2/alpha SMA, Cspg4/Neuronglial 2-NG2, CD146/ Melanoma cell adhesion molecule, CD248/Tumor Endothelial marker 1-TEM1) not restricted to MSC but also shared with pericytes (first identified by the French scientist Charles Rouget in 1873), vascular smooth muscle cells (VSMC) and perivascular fibroblasts [1–7]. Of note, the latter cell subset do not really fit the definition of pericyte because they are not embedded in the vascular basement membrane [8].

It is now generally accepted that a large pool of MSC is found essentially at the perivascular level and with morphology and marker expression profile similar to pericytes [5,9].  Several studies have shown that post-capillary venule pericytes from the bone marrow are able to differentiate into differentiated MSC such as osteoblasts and chondrocytes in vivo [10]. More recent genetic lineage-tracing experiments and single-cell RNA sequencing data has reinforced a close link between pericytes and MSC phenotypes, particularly in the CNS, a tissue where the highest density of pericytes has been found in the body. RNA profiling of mouse brain vasculature revealed a rather unique pool of perivascular cells made of a two pericyte clusters and three subsets of perivascular fibroblasts [11,12]. Pathway and gene ontology enrichment analyses revealed that Fibulin+ type I fibroblasts are the main subtype involved in ECM production and fibrosis. The type III (cell migration-inducing protein-CEMIP+ perivascular fibroblasts) showed robust expression of various growth factors, including VEGF-A. Interestingly, the type I to type II (potassium calcium-activated channel subfamily M alpha 1-KCNMA1+ fibroblasts) trajectory was continuous with pericyte type 2 suggesting a lineage from type I to type II to pericytes and consistent with a study in zebrafish demonstrating the stem cell potential of perivascular fibroblasts to differentiate into pericytes [13]. It was estimated that ten perivascular fibroblasts were present per intersegmented vessel but only less than 10% of these cells could differentiate into pericytes.  Garcia et al further discussed the possibility that type II perivascular fibroblasts in the brain probably represent an intermediate state exhibiting a transitional mural cell transcriptional phenotype [12].

            MSC have several different developmental origins as reviewed by Majeski [14]. The majority of the MSC/pericytes in the head region, including the CNS, are neural crest (NC) derived, as demonstrated in chick-quail chimeras carried out initially by the French scientist Nicole le Douarin and colleagues  [15]. In the peripheral nerves, NC will give rise to perineurial fibroblasts and Schwann cells [16]. More recently, two independent studies published in 2017 have suggested that brain pericytes could also be derived from mesoderm-derived myeloid progenitor cells [17,18]. Studies on the thymus demonstrated that perivascular MSC/pericytes are derived from the NC [19–21]. The origins of pericytes in the gut [22], lung [23] and liver [24,25]  have been mapped to the mesothelium although NC can give rise to MSC-like cells in the gut [26]. 

In the kidney, the metanephric mesenchyme of the intermediate mesoderm will give rise to nephrons (from the distal convoluted tubule to the podocytes) and also to all major stromal interstitial cells, including the pericytes, perivascular fibroblasts, VSMCs and mesangial cells [27]. a NC origin of a subset of perivascular fibroblasts of the kidney and contributing to fibrosis has been proposed from genetic lineage tracing experiments using the Schwann cell Myelin P Zero promoter-GFP/LacZ mice [28]. In the aorta MSC may have at least four different developmental origins, secondary heart field, NC, somites, and splanchnic mesoderm. This invasion of mesothelial cells occurs at about the same time as the appearance of primitive endothelial and hematopoietic progenitors within the splanchnopleura. The primitive endothelial cells (EC) within the splanchnopleura colonize the floor of the aorta and differentiate in situ to produce the vasculature of the body wall, kidney, visceral organs, and limbs [14]. This process of vasculogenesis involving PDGF high expression by EC is consistent with the notion that mesothelial-derived MSCs are localized to BM via the invasion of the vasculature.  Coronary vessels in the heart appear to have a similar development [29]. Mesothelial cells are known to undergo epithelial-to-mesenchymal transition (EMT) to delaminate and to migrate into the organs to produce their mesenchymal components. Interestingly, recent studies also point to a close ontogenic relationship between pericytes/VSMC and perivascular fibroblasts in many organs and supporting the current paradigm of such relationships in pathological settings for instance in the brain and lungs. The recent studies preach for the existence of a continuum of pericytes/perivascular MSC-fibroblasts cell phenotypes observed along vessels (and possibly nerves) and which suggest that these cells can (trans)differentiate into each other in conjunction with vessel/axonal/tissue remodeling. However, this interesting and promising paradigm requires further investigation.

            The close relationship of MSC and progenitors with the vasculature will endow them as a possible source of new cells for physiological turnover for the repair or regeneration of local lesions. The canonical and current scenario is that damage to any tissue would release the MSC from its perivascular niche, they will divide and secrete immunoregulatory and trophic factors. Different signaling mechanisms may govern MSC mobilization from the perivascular niche, detaching from the endothelial cues and invading the parenchyma in response to injuries.  This is exemplified by the importance of PDGF-B/PDGFRb which has been demonstrated in many organs such heart, lung, and gut [30].

            To date, we have a better idea into the embryonic origin of pericytes/perivascular MSC-fibroblasts in different organs but it is still critical to decipher the mechanisms governing their proliferation spreading along (as well evading) growing vessels in conjunction with angiogenesis. The capacity of these cells to circulate in the blood in various disease settings is of great and emerging importance from a clinical standpoint and including the identification of novel predictive soluble biomarkers of an ongoing pathological process in the tissues. Indeed, CD45−CD31−PDPN+ proinflammatory mesenchymal, or PRIME, cells have been identified in the blood from patients with rheumatoid arthritis and these cells shared features of inflammatory synovial fibroblasts and predicting inflammatory flares [31]. This line of future studies is of great importance in cancer and other chronic inflammatory diseases associated with infectious diseases. In the context of cancer, some studies relate to the capacity of NG2+ pericytes to give rise to mesenchymal tumors (i.e. osteosarcoma) [32].

new and amended references :

  1. Meirelles, L. da S.; Caplan, A.I.; Nardi, N.B. In Search of the in Vivo Identity of Mesenchymal Stem Cells. Stem Cells 2008, 26, 2287–2299, doi:10.1634/stemcells.2007-1122.
  2. Armulik, A.; Genove, G.; Betsholtz, C. Pericytes: Developmental, Physiological, and Pathological Perspectives, Problems, and Promises ». Developmental Cell 2011, 21, 193–215, doi:10.1016/j.devcel.2011.07.001.
  3. Bergers, G.; Song, S. The Role of Pericytes in Blood-Vessel Formation and Maintenance. Neuro-Oncology 2005, 7, 452–464, doi:10.1215/S1152851705000232.
  4. Meirelles, L. da S.; Chagastelles, P.C.; Nardi, N.B. Mesenchymal Stem Cells Reside in Virtually All Post-Natal Organs and Tissues. Journal of Cell Science 2006, 119, 2204–2213, doi:10.1242/jcs.02932.
  5. Crisan, M.; Yap, S.; Casteilla, L.; Chen, C.-W.; Corselli, M.; Park, T.S.; Andriolo, G.; Sun, B.; Zheng, B.; Zhang, L.; et al. A Perivascular Origin for Mesenchymal Stem Cells in Multiple Human Organs. Cell Stem Cell 2008, 3, 301–313, doi:10.1016/j.stem.2008.07.003.
  6. Duffield, J.S.; Lupher, M.; Thannickal, V.J.; Wynn, T.A. Host Responses in Tissue Repair and Fibrosis. Annu Rev Pathol 2013, 8, 241–276, doi:10.1146/annurev-pathol-020712-163930.
  7. El Agha, E.; Kramann, R.; Schneider, R.K.; Li, X.; Seeger, W.; Humphreys, B.D.; Bellusci, S. Mesenchymal Stem Cells in Fibrotic Disease. Cell Stem Cell 2017, 21, 166–177, doi:10.1016/j.stem.2017.07.011.
  8. Shaw, I.; Rider, S.; Mullins, J.; Hughes, J.; Peault, B. Pericytes in the Renal Vasculature: Roles in Health and Disease. Nat. Rev. Nephrol. 2018, 14, 521–534, doi:10.1038/s41581-018-0032-4.
  9. Hungerford, J.E.; Little, C.D. Developmental Biology of the Vascular Smooth Muscle Cell: Building a Multilayered Vessel Wall. J. Vasc. Res. 1999, 36, 2–27, doi:10.1159/000025622.
  10. Farrington-Rock, C.; Crofts, N.J.; Doherty, M.J.; Ashton, B.A.; Griffin-Jones, C.; Canfield, A.E. Chondrogenic and Adipogenic Potential of Microvascular Pericytes. Circulation 2004, 110, 2226–2232, doi:10.1161/01.CIR.0000144457.55518.E5.
  11. Vanlandewijck, M.; He, L.; Mae, M.A.A.; Andrae, J.; Ando, K.; Del Gaudio, F.; Nahar, K.; Lebouvier, T.; Lavina, B.; Gouveia, L.; et al. A Molecular Atlas of Cell Types and Zonation in the Brain Vasculature. Nature 2018, 554, 475-+, doi:10.1038/nature25739.
  12. Garcia, F.J.; Sun, N.; Lee, H.; Godlewski, B.; Mathys, H.; Galani, K.; Zhou, B.; Jiang, X.; Ng, A.P.; Mantero, J.; et al. Single-Cell Dissection of the Human Brain Vasculature. Nature 2022, 603, 893-+, doi:10.1038/s41586-022-04521-7.
  13. Rajan, A.M.; Ma, R.C.; Kocha, K.M.; Zhang, D.J.; Huang, P. Dual Function of Perivascular Fibroblasts in Vascular Stabilization in Zebrafish. PLoS Genet. 2020, 16, e1008800, doi:10.1371/journal.pgen.1008800.
  14. Majesky, M.W. Developmental Basis of Vascular Smooth Muscle Diversity. Arterioscler. Thromb. Vasc. Biol. 2007, 27, 1248–1258, doi:10.1161/ATVBAHA.107.141069.
  15. Etchevers, H.C.; Vincent, C.; Le Douarin, M.; Couly, G.F. The Cephalic Neural Crest Provides Pericytes and Smooth Muscle Cells to All Blood Vessels of the Face and Forebrain. Development 2001, 128, 1059–1068.
  16. Joseph, N.M.; Mukouyama, Y.S.; Mosher, J.T.; Jaegle, M.; Crone, S.A.; Dormand, E.L.; Lee, K.F.; Meijer, D.; Anderson, D.J.; Morrison, S.J. Neural Crest Stem Cells Undergo Multilineage Differentiation in Developing Peripheral Nerves to Generate Endoneurial Fibroblasts in Addition to Schwann Cells. Development 2004, 131, 5599–5612, doi:10.1242/dev.01429.
  17. Yamazaki, T.; Nalbandian, A.; Uchida, Y.; Li, W.; Arnold, T.D.; Kubota, Y.; Yamamoto, S.; Ema, M.; Mukouyama, Y. Tissue Myeloid Progenitors Differentiate into Pericytes through TGF-Beta Signaling in Developing Skin Vasculature. Cell Reports 2017, 18, 2991–3004, doi:10.1016/j.celrep.2017.02.069.
  18. Yamamoto, S.; Muramatsu, M.; Azuma, E.; Ikutani, M.; Nagai, Y.; Sagara, H.; Koo, B.-N.; Kita, S.; O’Donnell, E.; Osawa, T.; et al. A Subset of Cerebrovascular Pericytes Originates from Mature Macrophages in the Very Early Phase of Vascular Development in CNS. Sci Rep 2017, 7, 3855, doi:10.1038/s41598-017-03994-1.
  19. Jiang, X.B.; Rowitch, D.H.; Soriano, P.; McMahon, A.P.; Sucov, H.M. Fate of the Mammalian Cardiac Neural Crest. Development 2000, 127, 1607–1616.
  20. Foster, K.; Sheridan, J.; Veiga-Fernandes, H.; Roderick, K.; Pachnis, V.; Adams, R.; Blackburn, C.; Kioussis, D.; Coles, M. Contribution of Neural Crest-Derived Cells in the Embryonic and Adult Thymus ». The Journal of Immunology 2008, 180, 3183–3189, doi:10.4049/jimmunol.180.5.3183.
  21. Zachariah, M.A.; Cyster, J.G. Neural Crest-Derived Pericytes Promote Egress of Mature Thymocytes at the Corticomedullary Junction. Science 2010, 328, 1129–1135, doi:10.1126/science.1188222.
  22. Wilm, B.; Ipenberg, A.; Hastie, N.D.; Burch, J.B.E.; Bader, D.M. The Serosal Mesothelium Is a Major Source of Smooth Muscle Cells of the Gut Vasculature. Development 2005, 132, 5317–5328, doi:10.1242/dev.02141.
  23. Que, J.; Wilm, B.; Hasegawa, H.; Wang, F.; Bader, D.; Hogan, B.L.M. Mesothelium Contributes to Vascular Smooth Muscle and Mesenchyme during Lung Development. Proc. Natl. Acad. Sci. U. S. A. 2008, 105, 16626–16630, doi:10.1073/pnas.0808649105.
  24. Asahina, K.; Zhou, B.; Pu, W.T.; Tsukamoto, H. Septum Transversum-Derived Mesothelium Gives Rise to Hepatic Stellate Cells and Perivascular Mesenchymal Cells in Developing Mouse Liver. Hepatology 2011, 53, 983–995, doi:10.1002/hep.24119.
  25. Cassiman, D.; Libbrecht, L.; Desmet, V.; Denef, C.; Roskams, T. Hepatic Stellate Cell/Myofibroblast Subpopulations in Fibrotic Human and Rat Livers. J. Hepatol. 2002, 36, 200–209, doi:10.1016/S0168-8278(01)00260-4.
  26. Kruger, G.M.; Mosher, J.T.; Bixby, S.; Joseph, N.; Iwashita, T.; Morrison, S.J. Neural Crest Stem Cells Persist in the Adult Gut but Undergo Changes in Self-Renewal, Neuronal Subtype Potential, and Factor Responsiveness. Neuron 2002, 35, 657–669, doi:10.1016/S0896-6273(02)00827-9.
  27. Kobayashi, A.; Mugford, J.W.; Krautzberger, A.M.; Naiman, N.; Liao, J.; McMahon, A.P. Identification of a Multipotent Self-Renewing Stromal Progenitor Population during Mammalian Kidney Organogenesis. Stem Cell Rep. 2014, 3, 650–662, doi:10.1016/j.stemcr.2014.08.008.
  28. Asada, N.; Takase, M.; Nakamura, J.; Oguchi, A.; Asada, M.; Suzuki, N.; Yamarnura, K.; Nagoshi, N.; Shibata, S.; Rao, T.N.; et al. Dysfunction of Fibroblasts of Extrarenal Origin Underlies Renal Fibrosis and Renal Anemia in Mice. J. Clin. Invest. 2011, 121, 3981–3990, doi:10.1172/JCI57301.
  29. Dettman, R.W.; Denetclaw, W.; Ordahl, C.P.; Bristow, J. Common Epicardial Origin of Coronary Vascular Smooth Muscle, Perivascular Fibroblasts, and Intermyocardial Fibroblasts in the Avian Heart. Dev. Biol. 1998, 193, 169–181, doi:10.1006/dbio.1997.8801.
  30. Hellstrom, M.; Kalen, M.; Lindahl, P.; Abramsson, A.; Betsholtz, C. Role of PDGF-B and PDGFR-Beta in Recruitment of Vascular Smooth Muscle Cells and Pericytes during Embryonic Blood Vessel Formation in the Mouse. Development 1999, 126, 3047–3055.
  31. Orange, D.E.; Yao, V.; Sawicka, K.; Fak, J.; Frank, M.O.; Parveen, S.; Blachere, N.E.; Hale, C.; Zhang, F.; Raychaudhuri, S.; et al. RNA Identification of PRIME Cells Predicting Rheumatoid Arthritis Flares. N. Engl. J. Med. 2020, 383, 218–228, doi:10.1056/NEJMoa2004114.
  32. Sato, S.; Tang, Y.J.; Wei, Q.; Hirata, M.; Weng, A.; Han, I.; Okawa, A.; Takeda, S.; Whetstone, H.; Nadesan, P.; et al. Mesenchymal Tumors Can Derive from Ng2/Cspg4-Expressing Pericytes with Beta-Catenin Modulating the Neoplastic Phenotype. Cell Reports 2016, 16, 917–927, doi:10.1016/j.celrep.2016.06.058.